# Atherosclerosis and Inflammation: Insights from the Theory of General Pathological Processes

**DOI:** 10.3390/ijms24097910

**Published:** 2023-04-26

**Authors:** Evgenii Gusev, Alexey Sarapultsev

**Affiliations:** 1Institute of Immunology and Physiology, Ural Branch of the Russian Academy of Science, 620049 Ekaterinburg, Russia; a.sarapultsev@gmail.com; 2Russian-Chinese Education and Research Center of System Pathology, South Ural State University, 454080 Chelyabinsk, Russia

**Keywords:** aging, atherosclerosis, autoimmunity, cellular stress, endotheliosis, general pathological processes, polarization of macrophages and T-lymphocytes, systemic inflammation

## Abstract

Recent advances have greatly improved our understanding of the molecular mechanisms behind atherosclerosis pathogenesis. However, there is still a need to systematize this data from a general pathology perspective, particularly with regard to atherogenesis patterns in the context of both canonical and non-classical inflammation types. In this review, we analyze various typical phenomena and outcomes of cellular pro-inflammatory stress in atherosclerosis, as well as the role of endothelial dysfunction in local and systemic manifestations of low-grade inflammation. We also present the features of immune mechanisms in the development of productive inflammation in stable and unstable plaques, along with their similarities and differences compared to canonical inflammation. There are numerous factors that act as inducers of the inflammatory process in atherosclerosis, including vascular endothelium aging, metabolic dysfunctions, autoimmune, and in some cases, infectious damage factors. Life-critical complications of atherosclerosis, such as cardiogenic shock and severe strokes, are associated with the development of acute systemic hyperinflammation. Additionally, critical atherosclerotic ischemia of the lower extremities induces paracoagulation and the development of chronic systemic inflammation. Conversely, sepsis, other critical conditions, and severe systemic chronic diseases contribute to atherogenesis. In summary, atherosclerosis can be characterized as an independent form of inflammation, sharing similarities but also having fundamental differences from low-grade inflammation and various variants of canonical inflammation (classic vasculitis).

## 1. Introduction

The term ‘atherosclerosis’ comes from the Greek words ‘athero’, which means gruel or paste, and ‘sclerosis’, which means hardness. Atherosclerosis is a disease that is genetically intended for everyone who reaches a certain age, and its complications, primarily coronary heart disease and stroke are the main causes of death in developed countries [1,2]. There is a concerning trend of atherosclerosis development in young people [3,4]. Recently, significant advances have been made in understanding the molecular and cellular mechanisms of atherosclerosis. However, the systematization of these numerous data from the position of general pathology remains an urgent problem. Therefore, this review aims to study the main patterns of atherosclerosis in terms of the main typical pathological processes.

Aim and Objectives of the Review:

Aim: To elucidate the typical patterns of atherosclerosis pathogenesis as a unique variant of the general pathological process of inflammation, which exhibits distinctive features of both classical (canonical) and low-grade inflammation.

Objectives:Characterize atherosclerosis as a pressing issue in modern medicine and provide a general description of the various types of inflammation (Section 2);Demonstrate the pathogenetic role of cellular stress in atherosclerosis as an elementary functional component of general pathological processes (Section 3);Evaluate the importance of low-grade inflammation, endothelial dysfunction, and other associated pathogenesis links in atherosclerosis (Section 4);Systematize the primary typical immune mechanisms of inflammation development in stable and unstable atherosclerotic plaques (Section 5);Offer a comprehensive assessment of the inflammatory process in atherosclerosis (Section 6);Ascertain the relationship between atherosclerosis and systemic hyperinflammation (Section 7);Outline the main patterns of atherosclerosis pathogenesis from the perspective of general pathology and the theory of general pathological processes (Section 8).

## 2. General Characteristics of Atherosclerosis and Typical Inflammatory Processes Associated with Atherosclerosis

### 2.1. Definition

Atherosclerosis is a chronic inflammatory disease of elastic and musculoelastic arteries associated with the formation of atheromatous plaques (cholesterol) that cause stenosis or thrombosis (if the plaque is unstable).

### 2.2. General Patterns of Development of Atherosclerosis

The main pathogenetic factors of atherosclerosis are:Aging is associated with atherosclerosis, but it is not the only factor since early atherosclerosis and familial (genetic) variants of cholesterol metabolism disorders are known [5]. Pathoanatomical signs of the initial manifestations of atherosclerosis are also detected in children who died of other causes [6,7]. Therefore, preventing atherosclerosis in people aged 20–40 years is essential [8]. However, clinical manifestations of atherosclerosis are primarily observed in the elderly, associated with age-related changes in the metabolome, endotheliosis, immune dysfunction, shortening of telomeres, and other systemic changes in homeostasis [9,10]. Dyslipoproteinemia, dyslipidemia, hyperglycemia, and metabolic dysfunctions contribute not only to atherosclerosis but also to cell aging in a vicious circle [11];Atherosclerosis is linked to lipid metabolism disorders, mainly with the effect of atherogenic forms of lipoproteins on vessels, primarily modified (oxidized or methylated) low-density lipoproteins (LDL) [12];Atherosclerosis is associated with the immune system and inflammation [13,14,15]. However, “inflammation” is a broad concept uniting processes that are heterogeneous in their pathogenetic significance. This predetermines the need to specify “inflammation” in atherosclerosis from the standpoint of methodological approaches to general pathology.

### 2.3. Risk Factors for Atherosclerosis Development

Atherosclerosis is a complex multifactorial disease with various risk factors that contribute to its development. These risk factors can be broadly classified into several categories, including pro-inflammatory processes and metabolic dysfunctions, congenital factors, gender differences, lifestyle risk factors, disorders of intestinal microbiota, infections, and anatomical features.

Pro-inflammatory processes and metabolic dysfunctions in atherosclerosis are closely associated with obesity, metabolic syndrome, and type 2 diabetes [16,17,18]. These pathologies are currently considered to be the result of low-grade chronic inflammation [19,20], which belongs to the category of nonclassical inflammation (meta-inflammation, para-inflammation, or quasi-inflammation) [21,22,23,24]. Low-grade chronic systemic inflammation is considered to be the basis of the pathogenesis of most age-related diseases, including hypertension and atherosclerosis [25,26,27];Congenital risk factors for atherosclerosis are associated with polymorphisms of polygenes that control the development of atherosclerosis, either directly or indirectly (e.g., through epigenetic mechanisms) [28,29,30,31,32]. The potential number of genes and genetic combinations that can serve as risk factors for atherosclerosis is very large [33]. In addition, there are familial variants of early atherosclerosis associated with specific genetic anomalies, primarily in cholesterol metabolism [34,35];Gender differences play a role in atherosclerosis development. Women under the age of about 70 are less susceptible to atherosclerosis than men, partly due to the effect of estrogen on lipid metabolism [36,37,38];Lifestyle risk factors such as smoking, unhealthy diet, alcohol abuse, chronic psychosocial stress, sleep deprivation, physical inactivity, and air pollution are considered risk factors for the disease [38,39,40,41,42];Disorders of intestinal microbiota and an increase in intestinal permeability for toxic products of microbial metabolism also contribute to the progression of atherosclerosis [43,44,45];Although atherosclerosis is not an infectious disease, infections, especially those that invade arteries, can serve as additional factors in the onset and progression of atherosclerosis [46,47,48];Anatomical features of the arteries, such as bifurcations, which exhibit turbulent blood flow compared to laminar flow, contribute to the progression of atherosclerosis [43]. The turbulent flow activates and repositions endotheliocytes, increasing their permeability to large molecules, including LDL. Atherosclerotic stenosis causes the transition to turbulence in the oscillatory flow of blood in the arteries, leading to greater hydrodynamic instability in the flow [49];Arterial hypertension shares common pathogenic mechanisms with atherosclerosis, primarily endothelial dysfunction, and is considered the most prevalent modifiable risk factor for atherosclerosis [50].

### 2.4. The Main Stages in the Development of Atherosclerosis

#### 2.4.1. Morphology of the Normal Vascular Wall of the Arteries [51,52]

The morphology of the normal vascular wall of the arteries is well-established in the literature [51,52]. The inner wall of the arteries is composed of a monolayer of endothelial cells (ECs), which are connected with tight and gap junctions and are situated on a basement membrane. ECs are derived from mesenchymal cells and can be characterized as squamous epithelium. Their cytoplasm is thin, approximately 0.2–0.4 microns, and contains numerous transport vesicles that can form transendothelial channels. A dense layer of glycocalyx separates the ECs from the blood, and its thickness can reach several microns. Under normal physiological conditions, the endothelium is stable, slowly renewed, and does not have a significant amount of pro-inflammatory markers.

Beneath the endothelial lining is the intima, which is a layer of loose connective tissue primarily composed of collagen fibers, glycosaminoglycans of the main substance, and occasional stromal macrophages (Mf). The media is the next layer and is composed mainly of vascular smooth muscle cells (VSMCs). The outer layer of the arteries is called the adventitia, which is made up of fibrous structures and various connective tissue cells, including stromal Mf. The middle shell is sandwiched between the inner and outer elastic plates. The intima is located inside the inner elastic plate, and the adventitia is located outside the outer elastic plate. The microcirculatory network vasa vasorum provides oxygen and nutrients to the tunica media and tunica adventitia.

#### 2.4.2. The Main Stages in the Development of Atherosclerosis

The development of atherosclerosis, a common disease in developed countries that significantly impacts life expectancy, involves a sequence of milestone events [15,53,54,55,56]. The events can be summarized as follows:ECs activation of results in the impairment of the barrier function of the glycocalyx and weakened contact between ECs, which increases the transport of LDL through the endothelial wall into the intima. This leads to the deposition of lipids in the intima in the form of “lipid bands.” Additionally, ECs produce chemokines and express adhesive molecules that facilitate the transendothelial migration of monocytes and other leukocytes;Monocytes migrate into the intima of the arteries, where they take up oxidized LDL (oxLDL). These monocytes differentiate into inflammatory Mf that accumulate cholesterol and become foam cells. Various forms of programmed necrosis, including incomplete apoptosis of foam cells, leading to the formation of a necrotic atheromatous nucleus consisting of cholesterol and cell degradation products. The release of necrotic cell products enhances the inflammatory response of cells surrounding the atheromatous nucleus;Activated VSMCs undergo transdifferentiating and become foam cells, fibroblasts, fibromyocytes, endothelial, adipocyte, and osteoblast-like cells, cells with a phenotype similar to mesenchymal stem cells, and cells of an intermediate phenotype that participate in the formation of foam cells (approximately up to 50%), fibrous plaque structures, and arterial calcification. Fibrous structures form the lid of an atherosclerotic plaque, and plaque sprouts microvessels from adventitia and media that provide nourishment to plaque cells and their periphery. If the plaque becomes unstable, its cap can rupture and form a blood clot inside the vessel. Rupture-prone atherosclerotic plaques often have large lipid cores covered by a thin fibrous sheath (<60 µm). Lesions with these characteristics are often referred to as “vulnerable plaques.” On the contrary, plaques with limited lipid accumulation and thicker fibrous caps are often referred to as “stable plaques.”

From a pathogenetic perspective, atherosclerosis is an inflammatory disease that appears to be an inevitable ontogenetic disease associated with local and systemic changes in homeostasis (allostasis), especially in the state of the endothelium, the immune system, and lipid metabolism. However, what is meant by “inflammation” in the context of general pathology in this case?

### 2.5. Classical and Non-Classical Variants of Inflammation, Their Possible Relationship with Atherosclerosis

The involvement of pro-inflammatory mechanisms at the tissue level, including the formation of a cytokine network, is a hallmark of inflammation as well as some physiological processes and diseases traditionally not classified as inflammatory, such as cancer and schizophrenia [57,58]. However, not all inflammatory processes can be characterized as canonical inflammation, including low-grade inflammation and systemic hyperinflammation [59,60].

#### 2.5.1. Canonical Inflammation

Canonical inflammation is characterized by a focal reaction of microvessels with an exudative response and directed migration of leukocytes to the site of local damage. This reaction of the endothelium of postcapillary venules, known as endotheliitis, can only occur in vertebrates that have a network of lymphatic capillaries and blood microcirculatory units in various organs [61]. These processes are responsible for the five characteristic signs of inflammation known since ancient times, namely rubor, tumor, calor, dolore (Celsus) [62], and functio laesa (Galen) [63]. In contrast, invertebrates exhibit local inflammation of a nonclassical type, characterized by the accumulation of phagocytes in the damage zone but without a microvascular component.

The primary goal of local inflammation, whether classical or nonclassical, is to isolate and eliminate damaging agents. The systemic inflammatory response in classical inflammation performs a different task, primarily the resource provision of the focus of inflammation, and is characterized by a lower level of damaging factors and intensity of pro-inflammatory processes. Various variants of classical inflammation can develop in the focus of inflammation, depending on the severity of exudative vascular and cellular reactions. Productive inflammation, which is dominated by mononuclear leukocytes and inflammatory Mf formed from monocytes, is closest to atherosclerosis. However, atherosclerosis does not fully correspond to this variant of inflammation because the inflammatory role of the vasa vasorum is not clear, four local signs of inflammation formulated by Celsus are not present, and the reaction of the endothelium of the arteries can be defined as endotheliosis, which is a characteristic feature of low-grade inflammation.

#### 2.5.2. Low-Grade Local and Systemic Inflammation [59,60]

Low-grade inflammation, also known as parainflammation, is a type of non-classical inflammation characterized by the long-term effects of damaging factors, the absence of characteristic signs of an inflammatory focus (such as an inflammatory reaction of microvessels), delocalization of the process, insufficiency of barrier mechanisms, and its interconnection with tissue aging processes, metabolic factors of damage, or systemic changes in the endothelium (endotheliosis) [59,60]. This type of inflammation is evident in conditions such as morbid obesity, metabolic syndrome, type 2 diabetes mellitus, and age-related sarcopenia [64,65]. While atherosclerosis is associated with these processes, it does not fully correspond to this variant of inflammation in terms of the severity of inflammatory mechanisms and other features of the plaque, such as local accumulation of immunocytes and pro-inflammatory changes.

#### 2.5.3. High-Grade Systemic Inflammation (Hyperinflammation) [59,60]

This type of nonclassical inflammation is characterized by a severe systemic effect of damaging factors that lead to systemic microvascular endotheliitis, which is typical for a classical inflammatory focus. It also results in life-threatening microcirculatory disorders, intravascular activation of leukocytes, hemostasis, complements, and kallikrein-kinin systems. In chronic systemic inflammation, there may be a “cytokine storm” with latent microcirculatory disorders. The association of atherosclerosis with these variants of nonclassical inflammation is not well understood and requires further investigation. Section 7 briefly discusses this problem.

#### 2.5.4. Cellular Stress as a Link between Various General Pathological Processes Associated with Inflammation

The fundamental functional unit of all types of inflammation, as well as several other pathological and even extreme physiological processes, is pro-inflammatory cellular stress [60]. One distinct attribute of cellular stress is the development of an inflammatory receptor and secretory phenotype, which ultimately leads to alterations at the organ and organism level (Figure 1).

## 3. Cellular Stress as an Elementary Functional Component of the Inflammatory Process in Atherosclerosis

### 3.1. General Pathological Patterns of Cellular Stress

#### 3.1.1. Definition

Cellular pro-inflammatory stress (CS) can be defined as “a complex of interrelated universal and specific cell processes in response to the action of factors that cause actual or potential damage”.

#### 3.1.2. Tissue Pro-Inflammatory Stress, Typical Pathological Processes of Inflammation, and Their Difference from Clinical Definitions

Models of general pathological (typical pathological) processes reflect the most fundamental patterns of human pathology. They are not clinical definitions but rather the theoretical basis for the formation of the latter. Inflammation is the most extensive general pathological process. Recent studies of the molecular mechanisms of the pathogenesis of various diseases have significantly expanded the concept of “inflammation”. Moreover, as noted earlier, various pro-inflammatory mechanisms (intracellular signaling pathways, cytokine networks, and other factors) are involved in many physiological processes [59]. This circumstance highlights the need to differentiate “inflammation” into various alternative general pathological processes, separating them from physiological processes and defining tissue stress as their common basis, with CS as an elementary component of this basis. However, CS must be differentiated from another common concept, namely “cell pathology”, which overlooks the cell’s response to damage and ignores the physiological significance of cell damage and controlled death (apoptosis and programmed necrosis).

### 3.2. Universal Links of CS in Atherosclerosis

The universal components of CS are depicted in Figure 2, where we can observe that changes in the genome, mitochondria, endoplasmic reticulum (ER), and other cellular compartments, as well as the activation effects of pro-inflammatory mediators, cause not only specific but also universal reactions in many types of cells, making CS an integral and typical phenomenon (Figure 2) [59,60,61,67,68,69].

The components of cellular senescence can be classified into several categories [60]:Oxidative stress involves the accumulation of reactive oxygen species (ROS) and free radicals in cells, leading to modifications in protein post-translational processes and the synthesis of biomolecules essential for all CS processes. Excessive ROS production, which primarily occurs in mitochondria, can damage macromolecules and lead to vascular damage [70,71]. Endogenous antioxidants serve as a checkpoint against these adverse effects, but an imbalance in pro-oxidant/antioxidant mechanisms can lead to a sustained state of oxidative stress. There is a consensus that oxidative stress is associated with cellular aging and the pathological activation of ECs, Mf, and other atherosclerotic plaque cells [72,73,74];DNA damage response (DDR) is another important component of CS that allows cells to control genotoxic stress and ensure the accurate transmission of genetic information to subsequent generations. DNA damage accumulation can result in various DDR outcomes, including cell cycle arrest, cell aging, malignancy, and apoptosis [75]. More than 1000 proteins, such as nuclear chaperones (ubiquitin, nucleophosmin, and SUMO protein), nuclear protein kinases (ATM, ATR, DNA-PKcs, and Chk1/2), nucleases, polymerases, ligases, DNA glycosylases, and transcription factors (TF), including p53, are involved in the DDR process in human cells [59,76,77]. DDR’s primary function is to stop the cell cycle, facilitate DNA repair, and ensure cell survival [78]. Conversely, apoptosis is an extreme variant of DDR that prevents the transmission of genetic abnormalities to daughter cells. Overall, DDR plays a protective role in vascular cell aging, promotes cell proliferation, and prevents the accumulation of cells with impaired differentiation in the focus of atherosclerosis [79,80,81];Mitochondrial stress is a complex process that involves the mitochondrial unfolded protein response reaction (UPR), associated with the inducible synthesis of heat shock proteins (HSP). Mitochondria are the main producers of ATP and ROS. There are approximately 1500 proteins that function in human mitochondria, of which only 13 are encoded by mitochondrial DNA (mtDNA) [82]. These are the most important proteins of mitochondrial respiratory complexes. The mitochondrial proteome is adjusted to the functional state of its cell and depends on the action of activating and damaging factors on the mitochondrion itself. The relationship between mitochondria and the cell nucleus is a well-established phenomenon that occurs in response to mitochondrial dysfunction. In this case, the UPR mt includes multidirectional changes in the biosynthesis of various mitochondrial proteins: a decrease in potentially toxic proteins but an increase in the production and transport into mitochondria of chaperones capable of restoring damaged mitochondrial proteins [83]. Integrative mitochondrial stress is primarily associated with the activation and production of HSP and several kinases by ATF4 that integrate mitochondria into the CS system [84]. Mitochondrial stress serves to reduce organelle damage and dysfunction, but mitochondrial factors also contribute to apoptosis, programmed necrosis, cellular senescence, and other outcomes of CS [60]. Currently, there are a significant number of facts that prove the key, but ambiguous, significance of mitochondrial stress in ECs, Mf, GMCS, and other cells during the development of various stages of atherosclerosis [85,86,87,88,89,90];ER stress occurs when there is a disruption of ER integrity or accumulation of misfolded proteins within it. This initiates UPR ER, primarily in the form of UPR ER [91]. The UPR ER process serves to restore altered ER homeostasis with the following goals: (1) arresting the synthesis and excretion of secretory proteins from the cell; (2) increasing transcription of chaperones and other proteins involved in protein folding and maturation; (3) utilizing denatured proteins via the ER-associated degradation complex (ERAD) [92]. Thus, the UPR ER provides a coordinated response that contributes to overcoming impaired ER proteostasis. The prolonged or critical intensity of UPR ER, in turn, induces apoptosis in several ways, including Ca^2+^ release into the cytoplasm of the ER [91,93]. However, since ER stress also activates antiapoptotic pathways (anti-apoptotic proteins of the Bcl-2 family), apoptosis is not the only outcome of ER stress. The development of ER stress is not an independent process of other components of CS [94,95]. In atherosclerosis, the development of ER stress in ECs, Mf, and other cells of focus of atheromatosis is closely associated with oxidative and mitochondrial stress, especially with cholesterol accumulation in foam cells [96,97,98,99,100,101];HSP response: The inducible synthesis of HSPs is a highly conserved and evolutionarily ancient molecular response that cells use to cope with disturbances in their proteostasis [102]. As mentioned earlier, HSPs play a crucial role in maintaining the UPR, and they are also involved in regulating several key cellular processes [103]. In fact, HSPs are involved in most of the major pathogenetic processes that lead to atherosclerosis, including autoimmune reactions [104]. However, the limited production of HSPs in the arteries during aging is a pathogenic mechanism that contributes to the development of atherosclerosis [105];Autophagy: Autophagy is a critical part of many physiological and pathological processes, and its severity increases significantly during starvation and severe CS with pronounced activation of catabolic processes [106]. In CS, the ubiquitin-proteasome pathway, which is responsible for the utilization of altered proteins, is overloaded, and this may act as an additional mechanism to activate autophagy [91,107]. Additionally, many long-lived proteins, large protein aggregates, and individual organelles can only be recycled through the autophagy process, which involves lysosomes and numerous accessory protein factors. Mitophagy, in particular, is the only mechanism for physiological mitochondrial recycling [108]. Therefore, autophagy plays a crucial role in maintaining the balance between protein biosynthesis, organelle biogenesis, and organelle degradation. Furthermore, autophagy can also prevent cell apoptosis or necrosis by removing damaged mitochondria, protein complexes, and intracellular parasites. Autophagy is generally classified into three main types and is regulated at various levels through CS mechanisms [109]. However, the regulation and implementation of autophagy can be altered as cells age [110]. In cells involved in the pathogenesis of atherosclerosis, the intensity of autophagy/mitophagy tends to increase, and the disruption of this process contributes to atheromatosis and its complications [111,112,113]. Adequate autophagy, proportional to the degree of damage, is an adaptive process in atherosclerosis. However, disproportionate autophagy itself contributes to cell damage and dysfunction in the focus of atheromatosis [114];Inflammasomes are multimeric cytosolic protein complexes with sensory molecules that recognize microbial and endogenous molecular patterns associated with damage. During the assembly of the protein complex, these receptors bind to procaspase-1, which is then converted into caspase-1. This induces the processing of IL-1β and IL-18 and, under certain conditions, leads to the development of pyroptosis (a variant of programmed necrosis) [115]. The formation of inflammasomes is indicative of a relatively pronounced cellular stress (CS). In the pathogenesis of atherosclerosis, the key role is played by NLRP3 inflammasomes, which are activated by a wide variety of factors, including cellular decay products, reactive oxygen species (ROS), lysosomal proteinases, cholesterol and calcium phosphate crystals, numerous exogenous stimuli, and the release of mitochondrial DNA into the cytoplasm [116,117,118]. Disruption of control over the formation and utilization of inflammasomes may inappropriately amplify the inflammatory response, particularly in macrophages, thereby exacerbating the progression of atherosclerosis [119,120];Noncoding RNAs, including microRNAs and long noncoding RNAs, have been shown to modulate gene expression and play a role in the development of atherosclerosis [121,122,123,124,125]. Noncoding RNAs can regulate cellular processes involved in proteostasis and autophagy, as well as intercellular communication, by being incorporated into extracellular vesicles [126]. Recent developments have focused on the potential of using miRNAs as a therapeutic approach for treating atherosclerosis [127];Formation of stress granules: In the post-transcriptional stage, RNA-binding proteins play a key role in stress-induced regulation of the fate and function of various RNAs [128]. Protein-RNA complexes can form large granules (approximately 0.1–1.0 μm) in the cytoplasm of stressed cells, whose function is presumably to protect and store mRNA [129]. Additionally, due to their non-membrane characteristics, these structures can serve as a center where various cellular signaling pathways converge and exchange components [130]. Meanwhile, CS can induce the formation of gel-like structures in cells, including those involving amyloid and prion-like proteins [131]. The formation of these structures is dynamic; they quickly condense or dissolve, making them ideal for participating in emergency cellular adaptation to stress. However, when autophagy function is insufficient and insoluble granules are formed from these proteins, there is a decompensated increase in chronic stress and cell damage [132]. Immunohistochemical analysis of atherosclerotic plaques in mice revealed an increase in stress-granule-specific markers in intimal Mf and VSMCs in direct relation to disease progression [130,133];Formation of a network of signaling pathways in cellular stress: At the cellular level, stress development is mediated by complex epigenetic control programs and the interweaving of signaling pathways, the protein elements of which are constantly subjected to multiple post-translational modifications [134,135]. Various extracellular and intracellular stress signals can activate common collector-type protein kinases (e.g., MAPK, STAT, Akt, PI3K, PKC, ATM, ATR, AMPK, PKA, PKR, mTOR) and key universal factors TF associated with cellular stress (NF-κB, p53, AP-1, HIF, HSF, NRF2, ATF4) in different cell types [136,137,138,139,140,141,142,143,144,145,146]. The same signaling molecules can be activated and participate in various multidirectional processes; however, several pleiotropic TFs may display functional preferences. Thus, HIF-1 (hypoxia-inducible factor-1) plays a key role in the development of cellular stress in hypoxia [147]; HSF1 (heat shock factor 1) is involved in the production of heat shock proteins [148]; NRF2 (nuclear factor erythroid 2-related factor 2) triggers the production of antioxidants by a negative feedback mechanism in oxidative stress [149]; ATF4 (activating transcription factor 4) plays a decisive role in the development of UPRmt [84].

The dynamic network of signaling pathways integrates various elements of cellular CS into a single entity, including the receptor and secretory phenotype of pro-inflammatory cells. To date, almost all of these universal mechanisms of CS have been shown to be involved in the pathogenesis of atherosclerosis [53,150,151,152,153]. It has been demonstrated that in areas of arterial blood flow with impaired (turbulent) blood flow, many key transcription factors (TFs) are activated in ECs, including NRF2, HIF-1α, NF-κB, AP-1, and others [151]. Additionally, activation of Runx2 (Runt-related transcription factor 2) in pathologically activated VSMCs promotes its transformation into osteoblast-like cells, which leads to the calcification of atherosclerotic plaques [154].

Several transcription factors, such as NRF2, p53, ATF3, ATF4, HIF1α, HSF1, and MAPK, and several other stress protein kinases, have been shown to play a multidirectional role in the development of ferroptosis (a variant of programmed necrosis) in Mf within atherosclerotic plaques [155]. Lipid-activated TFs, peroxisome proliferator-activated receptors (PPARs), play a crucial role in regulating lipid and lipoprotein metabolism, glucose homeostasis, and inflammation in various cells. Both animal models of atherosclerosis and clinical studies strongly support the antiatherosclerotic role of PPAR alpha and gamma in vivo [156]. Furthermore, inhibitors of the NF-κB family have been shown to slow down the development of atherosclerosis by inhibiting TF activity [157]. Overall, the pro-inflammatory role of NF-κB is crucial for many functions and dysfunctions, including those in cardiovascular diseases [158].

Thus, in various cells involved in the pathogenesis of atherosclerosis, pro-inflammatory CS develops as a holistic phenomenon that includes a network of signaling pathways. A separate component of CS is the pro-inflammatory secretory phenotype, which allows the formation of information networks (primarily cytokine) of tissue stress.

### 3.3. Outcomes of CS and Their Importance in the Pathogenesis of Atherosclerosis

The genetically programmed purpose of CS is to ensure the safety and survival of cells, with the priority task of maintaining the body’s homeostasis. Therefore, different forms of programmed cell death, such as apoptosis and necrosis, cannot be considered inherently pathological. Cell aging is closely linked to CS, but it also has its own adaptive components, such as an alternative to tumor cell transformation. The massive pro-inflammatory transformation of cells, primarily within the immune system, is the basis for the development of various types of inflammation during significant tissue damage. Figure 3 outlines the main routes for the implementation of CS, which are directly related to the pathogenesis of atherosclerosis, and where the final outcomes of CS are apoptosis and cell necrosis.

Apoptosis and cell necrosis are important biological processes with different implications for tissue homeostasis and disease progression. Apoptosis is a natural process of programmed cell death that is crucial for normal cell turnover, proper development and functioning of the immune system, and the resolution of inflammation [159]. Dead cells produced by apoptosis are rapidly cleared by Mf without significant generation of damage-associated molecular patterns (DAMPs), which are signals that can activate immune cells and promote inflammation. Therefore, apoptosis is a “delicate” variant of cell death that maintains tissue integrity and limits inflammatory responses. However, when apoptosis exceeds the regenerative capacity of the organ, it can contribute to tissue atrophy and subsequent fibrosis [160]. Apoptosis can be induced by various signaling pathways, which can be divided into extrinsic and intrinsic pathways, depending on their dependence on caspase activation. External pro-apoptotic signals act on TNF family cytokine receptors, primarily the Fas receptor (CD95). The main intrinsic pathway of apoptosis involves the release of pro-apoptotic molecules from mitochondria, such as cytochrome C, which activates caspase 9 and other pro-apoptotic caspases [161]. The mitochondrial response is controlled by a balance between pro-apoptotic and anti-apoptotic proteins of the Bcl-2 family, IAP, and other factors [162,163]. Therefore, the development of apoptosis is the result of a complex interplay between different signaling pathways that determine the balance between pro- and anti-apoptotic factors.

In the context of atherosclerosis, increased apoptosis can have both beneficial and detrimental effects, depending on the cell type and stage of the disease. For example, endothelial cell apoptosis can initiate atherosclerosis [164]. Apoptosis in Mf and foam cells is associated with ER stress, which can lead to an apoptotic response mediated by both Fas and mitochondrial pathways [165]. Apoptotic cell death is an important feature of atherosclerotic plaques, but its impact on plaque stability and inflammation is complex. In the late stages of atherosclerosis, a high level of VSMCs apoptosis can cause the fibrous cap to become smaller and thinner, leading to the formation of unstable plaques [166]. However, adequate uptake of apoptotic cells by Mf, also known as efferocytosis, is important to prevent the progression of atherosclerosis [167,168]. Efferocytosis is mainly carried out by functionally polarized Mf in the M2 direction, which has moderate pro-inflammatory activity and promotes tissue fibrosis, as opposed to the more pro-inflammatory M1 Mf [169,170]. When efferocytosis fails, incomplete apoptosis or secondary necrosis can occur, resulting in the release of DAMPs and the activation of inflammatory responses, including the Mf polarization toward M1 and the inhibition of efferocytosis by the CD47 receptor [169,171]. This creates a vicious cycle that perpetuates inflammation and incomplete apoptosis, making the atherosclerotic plaque unstable.

In addition to incomplete apoptosis in an unstable atherosclerotic plaque, other variants of programmed necrosis can occur, such as necroptosis (of Mf and VSMCs), pyroptosis (a final function of inflammasomes), ferroptosis (fatal iron-dependent lipid peroxidation), and autophagic death of foam cells (uncontrolled autophagy) [165]. It has become clear that neutrophils are involved in the pathological process in various stages of atherosclerosis (from the primary activation of the arterial endothelium and the formation of an unstable plaque) [172]. The neutrophil-specific cellular program (including the necrosis variant), characterized by the extracellular release of cobweb-like DNA structures called NET (NETosis), plays a significant role in tissue damage [172].

As noted in Section 2, transdifferentiation is a common consequence of the development of CS, particularly for VSMCs. These cells can differentiate into atypical cell populations with various functions that can have both adaptive and maladaptive significance.

The aging of cells is a result of the stochastic accumulation of damage in biomolecules such as the genome, transcriptome, and proteome, which are vital for proper cellular function. These changes can lead to the appearance of CS with the accumulation of ROS, cell cycle blockade, and the formation of cellular and tissue allostasis [173,174]. Cellular senescence is characterized by a state of proliferation arrest, in which cells remain metabolically active and secrete a range of pro-inflammatory and proteolytic factors and other components of the aging-associated secretory phenotype (SASP) [175]. Aging and pro-inflammatory SASP cells can form a vicious pathogenetic circle involved in the formation of tissue allostasis in aging tissue [176].

The aging of ECs and VSMCs is closely associated with vascular diseases such as hypertension and atherosclerosis [177,178]. Finally, aging Mf and other immunocytes that have accumulated genetic mutations can form populations of dysfunctional cells in atherosclerosis [9,179].

### 3.4. Triggers of Cellular Stress in Atherosclerosis

The development of atherosclerosis involves various factors that trigger cellular stress (CS). These factors can be classified into the following categories:Damage to macromolecules: Gradual accumulation of damage to the genome and proteome during aging, oxidative stress, and intracellular cholesterol accumulation during the formation of foam cells can cause damage to cells and extracellular matrix. This damage is recognized by CS sensors [180,181];Changes in homeostasis parameters: Several parameters of homeostasis can induce CS in atherosclerosis. These include an increase in osmotic [182,183] and hydrostatic pressure [183], changes in calcium concentrations in cells [184], and a decrease in ATP and oxygen concentrations in cells (hypoxia) [185,186];Lipotoxicity factors: Excess saturated free fatty acids (FFA), diacylglycerol, ceramides, modified carnitine, non-esterified cholesterol, and other hydrophobic molecules can damage mitochondria and other cellular structures [60,187,188,189];Recognition of PAMPs and DAMPs through PRRs: Pathogen-associated molecular patterns (PAMPs) and DAMPs recognized by pattern recognition receptors (PRRs) directly associate with inflammatory cells such as immunocytes, epitheliocytes, connective tissue cells, and endotheliocytes [190]. Cells can quickly enter a state of stress and realize their pro-inflammatory and immunocompetent functions. Toll-like receptors (TLRs) play a critical role in atherosclerosis [191]. TLRs contribute to the inflammatory Mf transformation and other cells in atheromatosis due to the release of significant amounts of DAMPs from necrotic cells [192,193]. TLRs can also be involved in earlier stages of atherosclerosis, including the development of endotheliosis, which is characteristic of low-grade inflammation and cardiovascular diseases [194,195]. Various stress cells can serve as a source of moderate amounts of DAMPs and HSP that interact with TLRs. Saturated FFA associated with blood plasma protein (FetA) may activate TLR4 [196], but the role of this mechanism in the pathogenesis of atherosclerosis is still unclear. TLR 4, expressed on various immunocytes, endotheliocytes and platelets, is the main receptor for bacterial endotoxin (LPS). Morbid obesity and systemic effects of lipotoxicity factors can lead to metabolic endotoxemia (an increase in LPS in the blood, approximately 1.5–2 times) as a result of impaired intestinal barrier function [197,198]. An increase in PAMPs in the blood can also be observed in infections indirectly associated with the development of atherosclerosis [199];Antibodies and T cell receptors (TCRs) recognize antigens, leading to T cell activation and interactions primarily with antigen-presenting cells. In atherosclerosis, these pathways of activation may be associated with the development of an auto-immune response to modified lipoproteins and other autoantigens [200,201];Excitotoxicity, in the narrow sense, refers to the toxic effect of high doses of glutamate on neurons, other neurotransmitters, and their catabolism products [202]. In the broad sense, it denotes the pathological hyperactivation of cells by various regulatory molecules, mainly pro-inflammatory cytokines such as TNF-α and IL-1β [59]. The phenomenon of cytokine-induced excitotoxicity contributes to the contagious development of chronic systemic inflammation. The formation of various variants of the local cytokine network in atherosclerosis is also an established fact [203,204];Scavenger receptors (SRs) are a family of molecular receptors that includes more than 30 members [170,205]. Although they do not share structural homology or genetic origin, they are united by common functional properties [206]. Structurally homologous receptors form separate SR classes (A-L) within this family. SRs are primarily located on the surface of stromal Mf but also on ECs and other cells, functioning at the intersection of metabolism, immunity, and inflammation. In particular, SRs are the primary sensors of modified LDLs. Another common feature of SRs is their ability to form functional clusters with Toll-like receptors (TLRs), integrins, and other receptors on cell membranes, modulating their activity in relation to the development of cardiovascular disease [206]. Due to this feature, SRs are involved not only in pathology but also in the implementation of physiological processes, providing clearance of abnormal metabolic products, relatively low concentrations of PAMPs and DAMPs, and apoptotic and abnormal cells. Moreover, in atherosclerosis, SRs can be involved in both adaptive and maladaptive processes.

Most SRs are involved in the processes of atherosclerosis. SR-E1 (LOX-1) is involved in endothelial cell activation and disruption of the endothelial barrier function, as well as in the activation of Mf and VSMCs [207,208]. SR-A1 (CD204) plays a key role in the uptake of modified LDL by Mf and VSMCs [209], while SR-B2 (CD36) plays a key role in the activation of lipoprotein-dependent Mf [210], and SR-J1 (RAGE) promotes negative activation of ECs and Mf [211]. At the same time, SR-J1 recognizes advanced glycation end products (AGEs), some DAMPs, and ROS-modified endogenous proteins (Table 1). SR-A1, SR-L1, and many other SRs on liver cells absorb modified LDLs, thus preventing their accumulation in the bloodstream and having an opposite effect on the development of atherosclerosis [170]. The nature of the course of atherosclerosis significantly depends on the morphological and functional features of the Mf formed and their ratio. Therefore, Mf with characteristic features of M2 (CD206 high, CD163 high/SR-E3, SR-I1) is mainly located in the periphery of atherosclerotic plaques in the annulus fibrosus zone [212].

In contrast, M1 Mf-expressing SR-B2 and TLR, with a high pro-inflammatory and procoagulant potential, are more concentrated in the central region and shoulder of plaques directed into the lumen of the vessel [212]. It is with these types of Mf that the increase in pro-inflammatory mechanisms, the involvement of the blood coagulation system, the formation of atheromas, and other complications of atherosclerosis are associated [212,213]. Additionally, plaque core cells are more saturated with cholesterol than peripheral Mf while maintaining a relatively high level of expression of the main oxLDL scavenger, SR-A1 (CD204) [212].

The development of oxidative stress in the endothelium and M1 Mf contributes to an additional modification of LDL. In contrast, the predominance of M2 Mf promotes scarring of the focus of atherosclerosis and restoration of the endothelial lining of the vessel. At various stages of atherosclerosis, pro-inflammatory CD8^+^ and CD4^+^ T cells, as well as natural killer (NK) cells, can be involved in the process [263], and their migration can be facilitated by the soluble form of SR-G1 secreted by Mf (chemokine CXCL16) [264]. In turn, the function of SR-B1 in the development of atherosclerosis can be ambiguous. Therefore, on the one hand, SR-B1 promotes the transendothelial transport of LDL in the arteries [265], but on the other hand, it has anti-inflammatory and anti-atherogenic functions in the focus of atheromatosis [266].

As noted above, the mechanisms of the formation of an unstable plaque are largely associated with the failure of the efferocytosis process. Stromal and M2 Mf recognize apoptosis products primarily through SRs that bind spectrin and phosphatidylserine fixed on the cell membrane, HSP, complement fragment C1q, and some other markers of apoptosis (Table 1).

In general, numerous SRs actively manifest their role in atherosclerosis processes, from the transition from physiology to pathology to the development of various types of inflammation in vascular atheromas. Therefore, the orientation of these functions can have both adaptive and maladaptive significance, depending on the type of receptor, stage, and other characteristics of the pathological process.

### 3.5. Summary

Atherosclerosis is a disease characterized by low-grade inflammation, which is induced by pro-inflammatory CS. The accumulation of cells with a pro-inflammatory phenotype in atherosclerotic plaque leads to a process similar to canonical inflammation. Pro-inflammatory CS includes mitochondrial and ER stress, oxidative stress, formation of a pro-inflammatory secretory phenotype, inflammasomes, autophagy processes, and HSPs and noncoding RNAs. The main inducers of CS are the gradual accumulation of cellular damage during aging and the effect of aberrant metabolome factors, such as modified LDL, saturated FFA, and AGE, on cells. The direction and degree of inflammation in atheromas strongly depend on efferocytosis, cell necrosis, and DAMPs formation, which can lead to unstable plaque. The role of various types of SRs is crucial in controlling the development of CS.

## 4. Low-Grade Inflammation, Endothelial Dysfunction, and Associated Non-Immune Links in the Pathogenesis of Atherosclerosis

Typical pathological processes fall under the categories of general pathology and pathological physiology. These processes serve as the fundamental and regular manifestations of the pathogenesis of different diseases and provide the theoretical foundation for clinical definitions, including nosology and syndromes.

Atherosclerosis, one such disease, is characterized by two distinct pathological processes, low-grade inflammation and canonical inflammation. A common factor in the development of atherosclerosis is “inflamm-aging,” which can be considered low-grade systemic chronic inflammation in certain cases [9,267,268]. Common symptoms of low-grade systemic inflammation include morbid obesity, allostatic insulin resistance, endotheliosis (often with essential hypertension [269], relatively low manifestations of acute phase response and hypercytokinemia, hypercholesterolemia, hyperlipidemia, and increased blood concentrations of atherogenic forms of lipoproteins and AGE [59,64,270,271,272,273]. As atherosclerosis progresses, it can result in the development of metabolic syndrome (type 2 diabetes) alongside hepatosis and sarcopenia [274,275].

From this perspective, the initial stages of atherosclerosis are characterized by a localized increase in endotheliosis in the arterial wall and systemic manifestations of low-grade inflammation. This contributes to metabolic dysfunctions and lipotoxicity, which can be exacerbated by genetic predisposition, lifestyle risk factors, infections, increased intestinal permeability to PAMPs and toxins, chronic psychoemotional stress, environmental factors, and local turbulence of arterial blood flow. As atherogenesis progresses, local manifestations of low-grade inflammation can transform into classical inflammation (for instance, in the progression of nonalcoholic fatty liver disease and diabetic kidney disease) [276,277]. While renal glomerular pathology in diabetes may be complicated by fibrinous inflammation, which is a type of classical-type exudative destructive inflammation, a similar trend is typical for atherosclerosis, although the growing signs of arterial wall pathology do not fully align with the canons of classical inflammation.

### 4.1. Typical Conditions and Manifestations of the Initial (Clinically Latent) Phenomena of Atherosclerosis

The concept of ‘inflamm-aging’ [278] is quite broad and encompasses not only the pathology related to low-grade inflammation but also the manifestations of pro-inflammatory tissue stress, which are necessary for the adaptation of the body to age-related changes in homeostasis (allostasis). Consequently, it is not surprising that in aging, there is a natural tendency towards an acute phase response of the liver and hypercytokinemia [279].

Our data shows that 18% of individuals over the age of 65 years without signs of acute and severe chronic diseases, including metabolic syndrome and morbid obesity, have signs of a systemic inflammatory response [66]. However, the increase in the levels of C-reactive protein (CRP) and pro-inflammatory cytokines (TNF-α, IL-6, IL-8) in these patients does not exceed the upper value of the reference range by more than two-fold. In general, the accumulation of genomic mutations in cells, oxidative stress, mitochondrial dysfunction, and other manifestations of cellular and tissue stress are also typical of aging not complicated by low-grade inflammation and associated pathologies (morbid obesity, metabolic syndrome, type 2 diabetes). Low-grade inflammation should be considered a specific stage in the development of ‘inflamm-aging.’

An important issue to address is how to differentiate between low-grade inflammation, chronic systemic hyperinflammation, and manifestations of the systemic inflammatory response in classical types of chronic inflammation, such as autoimmune and infectious diseases. Pro-inflammatory tissue stress, as discussed in Section 3.3, is also characteristic of tumor diseases. Therefore, it is crucial to distinguish low-grade inflammation from other forms of inflammation to understand its role in the development of age-related pathologies.

The concept of ‘inflamm-aging’ involves both adaptive and maladaptive mechanisms, and age-related pro-inflammatory status can only be considered age-related pathology in specific cases. The transition from “age norm” to “age pathology” is not a discrete process and has a “grey” marginal zone of prepathological, latent changes. Local and systemic pro-atherogenic changes that occur before obvious signs of lipid deposition in the arterial intima represent the marginal stage of atherogenesis (pre-atherosclerosis).

In addition, altered hemodynamics in the arteries during the first decade of atherogenesis may also be an independent factor of mitochondrial and oxidative stress in ECs, even before the morphological manifestations of lipid deposition [280]. However, the presence of lipid bands in the arterial intima indicates the development of the initial (subclinical) stage of atherosclerosis.

The main characteristic of these and subsequent stages of atheromatosis is the increase in signs of endothelial dysfunction. This dysfunction, or endotheliosis, is characterized by less pronounced tissue stress in ECs, in contrast to microvascular reactions in the focus of canonical inflammation [60].

### 4.2. Endotheliosis as a Key Phenomenon of Age-Related Changes and Age-Related Vascular Pathologies

In healthy arteries, the endothelium is generally in an inactive state with regard to tissue stress, unlike other tissues, such as covering tissues, lymphoid organs, and working muscles [59]. However, internal and external damaging factors and changes in regulatory mechanisms within ECs can lead to the development of chronic CS and endothelial dysfunction, which is referred to as endotheliosis. This condition is a key factor in the pathogenesis of various cardiovascular diseases [74]. The development of atherosclerosis, starting from its pre-atherosclerotic stages, is closely linked to the pro-inflammatory activation of ECs, leading to vascular tone release, structural remodeling, inflammation, and atherogenesis regulation. The balance between these factors is critical for maintaining vascular homeostasis, and disturbances in this balance due to endotheliosis can lead to the development of vascular pathologies [74]. Several typical phenomena of endotheliosis can be identified, and it is important to note that normal interactions between ECs, blood plasma cells and proteins, other arterial cells, and the subendothelial extracellular matrix are inevitably disrupted in this condition. Therefore, the dysfunction of ECs cannot be isolated a priori when analyzing these phenomena [281].

#### 4.2.1. Glycocalyx Pathology

The endothelial glycocalyx is a specialized extracellular matrix that covers the apical side of ECs and extends into the vascular lumen. Its composition, consisting of proteoglycans, glycosaminoglycans, and glycoproteins, has been extensively studied. The glycocalyx was once thought to be a passive physical barrier but is now known to be a multifunctional and dynamic structure involved in several vascular processes, such as vascular permeability, inflammation, thrombosis, mechanotransduction, and cytokine signaling [282,283].

Inflammation mediators such as cytokines and chemoattractants can cause degradation of the glycocalyx in different variants of inflammation, affecting arterioles, capillaries, and venules [284]. ROS, proteases, and glycosidases can directly damage the endothelial glycocalyx, which can also be produced by activated ECs. Moreover, ROS can enhance glycocalyx proteolysis by inactivating endogenous protease inhibitors [284,285,286]. Thus, a pathological feedback loop can occur between glycocalyx depletion and pro-inflammatory stress in ECs [286].

The barrier function of the endothelial glycocalyx deteriorates with age, preceding vascular dysfunction [287,288]. The degradation of the glycocalyx in various vessels is associated with negative local and systemic dynamics in the progression of atherosclerosis [289]. In oral submucosa capillaries, the degree of reduction in glycocalyx thickness correlates with the severity of coronary atherosclerosis [290]. Hypertension and arterial wall stiffness associated with aging can cause glycocalyx degradation in a pathological feedback loop, leading to increased endothelial dysfunction and vascular disease progression [291].

#### 4.2.2. Changes in Nitric Oxide Production

The decrease in nitric oxide (NO) production is a characteristic difference between endotheliosis and microvascular endotheliitis in classical inflammation. This decrease is primarily due to a decrease in constitutive endothelial NO synthase (eNOS) activity and reduced NO bioavailability [59]. In endotheliosis, there are insufficient pro-inflammatory stimuli to activate inducible NO synthase (iNOS), which is responsible for NO hyperproduction. iNOS promotes vasodilation of the arterioles and the exudative response of the microvessels while simultaneously limiting pro-inflammatory processes due to the regulation of negative feedback [292,293]. Therefore, in endotheliosis, the limitation of NO bioavailability contributes to vasoconstriction, thrombinemia, and the maintenance of a moderate pro-inflammatory state of ECs, which is milder than classical inflammation. This reduction in NO production, along with the simultaneous hyperproduction of other ROS, characterizes the development of various stages of atherosclerosis [294,295,296].

Elevated oxLDL concentrations in the blood microcirculation zone, along with additional stimuli for inflammation development, lead to sustained activation of SR-E1 (LOX-1) and subsequent activation of NF-kB [297]. This, in turn, can activate iNOS, leading to more pronounced oxidative stress with the formation of more reactive peroxynitrite from NO and the accelerated apoptosis of ECs [297]. A similar mechanism may occur in atherosclerosis. However, more specific evidence is needed at the level of the endothelium of large arteries, not only microvessels and venules, to confirm this hypothesis.

#### 4.2.3. Hyperproduction of Vascular Endothelial Growth Factors (VEGF)

One important characteristic of ECs and the cells surrounding them is their ability to produce growth factors, primarily from the VEGF family. These growth factors play a critical role in promoting vascular regeneration and the survival of ECs in extreme conditions. Most growth factors, including insulin, activate G-proteins and the PI3K/AKT/mTOR signaling pathway, which is associated with cell growth and anabolism. However, growth factors can also act through more specialized KC signaling pathways, which are unique to specific growth factors [59,298,299].

In particular, increased VEGF signaling has been shown to prevent age-related capillary loss, improve perfusion and organ function, and increase longevity in mice [300]. However, a chronic increase in VEGF-A production can lead to a more severe pro-inflammatory response, negatively affecting myocardial function in the aging heart [301]. Additionally, an increase in VEGF-A concentration can cause oxidative stress and multiple age-related eye diseases, such as cataracts, neovascular, and nonexudative pathologies [302].

In the development of atherosclerosis, the production of VEGFs can be induced by the TF family HIF, both as a result of hypoxia and hypoxia-independent signaling pathways such as TNF-α, LPS, and ROS/NF-κB/HIF/VEGFs [150,303]. This suggests that the strong inducers of VEGF and other growth factor production in ECs are mitochondrial and oxidative stress [304]. Additionally, oxLDL can activate NF-κB/HIF/VEGFs in ECs by acting on SR-E1 receptors (LOX-1) [233].

The effects of VEGFs on the development of atherosclerosis are complex and diverse. These effects have been described in detail in recent reviews, including [150,305,306,307]. VEGFs act on different cell types and are involved not only in angiogenesis and CS development but also in the regulation of lipid metabolism, glucose metabolism, and insulin sensitivity. Furthermore, the polymorphism of VEGF genes plays a significant role in predisposition to atherosclerosis and other cardiovascular diseases [308].

The VEGF family includes several members that play a role in atherosclerosis, with VEGF-A being a prominent marker [309]. VEGF-A has both beneficial and detrimental effects related to atherosclerosis. On the one hand, it protects ECs by stimulating the expression of antiapoptotic proteins and promoting NO synthesis by eNOS [310]. On the other hand, VEGF-A can prevent the repair of endothelial damage, contributing to atherogenesis and promoting monocyte adhesion, transendothelial migration, and activation [150].

Interestingly, the anti-atherogenic effects of VEGFs may be mediated through the regulation of arterial wall lymphatic drainage. Studies in mice suggest that a disruption of lymphatic function prior to the induction of atherosclerosis can contribute to the development of the disease, which is associated with a defect in the drainage function of the collecting lymphatic vessels. Systemic injections of VEGF-C can restore lymphatic transport in these mice [311].

#### 4.2.4. The Role of Noncoding RNAs

Endothelial dysfunction is associated with various epigenetic mechanisms that are regulated by noncoding micro (mi) and long (lnc) RNAs [312]. It is worth noting that a single miRNA can inhibit the translation of hundreds of mRNAs, and collectively, noncoding RNAs can affect a wide range of processes, including angiogenesis, apoptosis, proliferation, and migration in vascular cells [313].

In this study, we will examine several examples of how noncoding RNAs affect the functions of ECs. For instance, the MEG3 lncRNA was found to suppress proliferation and induce apoptosis in vascular ECs [314]. Furthermore, the regulatory pathway of RNA MEG3/miR-223/NLRP3 limits the activation of the NLRP3 inflammasome and the development of pyroptosis in ECs [315]. Additionally, the pathogenic role of lncRNA MALAT1 in the development of endothelial dysfunction in diabetes and aging was described [316,317]. LncRNA H19 was shown to prevent the endothelial-mesenchymal transition in diabetic retinopathy [318]. The lncRNA TUG1 and miR-148b contribute to the expression of the IGF2 protein, which regulates proliferation and apoptosis in cells treated with oxLDL [319]. Overexpression of lncRNA SENCR regulates proliferation and migration in the HUVEC endothelial cell line [320]. The lncRNA GATA6-AS has been shown to be involved in the response of ECs to hypoxia [321]. Increased lncRNA GAS5 expression and decreased miR-26a expression have been reported in plasma from patients with atherosclerosis in oxLDL-activated human aortic ECs [322]. LncRNA and miR-320a regulate SR-E1 (LOX-1) expression in oxLDL-activated HUVEC cells [319]. MiR-638 has been shown to negatively regulate the AKT/mTOR signaling pathway and therefore inhibit proliferation and activate apoptosis of ECs treated with human aortic oxLDL [323]. Overexpression of lncRNA FENDRR was found to regulate VEGF-A and promote apoptosis by targeting miR-126 [324]. The lncRNA ROR enhances NF-κB activation and the JAK1/STAT3 signaling pathway in EC, promoting the survival of these cells [325]. Overexpression of miR-134 suppresses the ability of ECs to grow and migrate [326]. An increased expression of RNCR3 lncRNA has been found in human aortic atherosclerotic lesions [327]. In ECs, miR-5680 inhibits AMPK phosphorylation, thus regulating the development of KS [328]. The NORAD lncRNA is activated in response to DNA damage and maintains mitosis stability and genome integrity [329]. Overexpression of lncRNA p21 in ECs increases apoptosis and expression of SR-E1 (LOX-1) [330]. LncRNA HIF1A-AS1 stimulates apoptosis and decreases the proliferative activity of ECs [331]. A study by Song et al. showed that coronary atherosclerosis could be prevented by reducing lncRNA ANRIL expression [332]. Inhibition of the lncRNA XIST/miR-320/NOD2 regulatory pathway promotes the survival of oxLDL-activated ECs [314].

MiR-155 is an extensively researched miRNA that is linked to inflammation. It plays a crucial role in regulating eNOS levels and disrupting EC-dependent vasorelaxation [333]. Conversely, miR-103 promotes endothelial inflammation and increases the adhesion of monocytes to ECs [333]. Activated neutrophils can deliver miR-155 in microvesicles to endothelial cells (ECs) in arterial regions prone to atherosclerosis, thereby enhancing EC activation and monocyte adhesion to ECs [334].

During plaque formation in the aorta, ECs export miR-92 in extracellular vesicles to Mf. As a result, Mfs alter their phenotype to pro-atherosclerotic [335]. MiR-92 is, therefore, a potential target for regulating vascular disease. The complex interplay between miRNA, lncRNA, and other regulatory factors controls the development and resolution of pro-inflammatory CS, including ECs. However, prolonged exposure to damaging factors can stabilize pathological changes at the cellular, tissue, and systemic levels, leading to the formation of stable allostasis and promoting inflammation at the local level, such as in arteries prone to atherosclerosis.

#### 4.2.5. Disruption of the Endothelial Barrier, Transendothelial Migration of Leukocytes in the Arteries

The process of cell migration across the endothelial barrier involves several fundamental mechanisms. First, there is ECs activation, followed by the adhesion of migrating cells to the endothelium and glycocalyx density reduction. Then, there is the dissociation of tight contacts between ECs, leading to the formation of gaps between them, which allows chemokines produced by ECs and their neighboring cells to direct the migration of leucocytes through the gaps formed via the paracellular pathway. Simultaneously, the paracellular pathway predominantly over the migration of leukocytes through the endothelial cell layer (transcellular pathway), which also necessitates prior adhesion of migrating cells to the endothelium [336,337].

The adhesion process includes several main steps. The first step is dynamic contact or “rolling”, which is determined by the inducible expression of E- and P-selectin receptors (CD62E/P) on ECs. The ligands for these receptors are leukocyte mucins, which are certain glycocalyx structures. P-selectins can be rapidly released from the Weibel-Palade body upon activation of ECs, which leads to the pre-activation of migrating leukocytes. Leukocyte (L) selectins (CD62L), which interact with endothelial mucins, also contribute to the rolling and pre-activation of migrating cells [336,337].

The second step is the formation of a sustained activation contact between cells or “arrest”, which involves ICAM-1 (CD54) and VCAM-1 (CD106) ECs. The main ligands of ICAM-1 are leucocyte β2-integrins (CD18), including LFA-1 (CD11a/CD18) and Mac-1 (CD11b/CD18). In turn, the main ligand of VCAM-1 is the β1-integrin (CD29), specifically VLA-4 (CD49b/CD29). These contacts are stimulated by several pro-inflammatory stimuli [336,338,339].

The third step is the disruption of contacts between ECs, which provides a paracellular pathway for leukocyte migration. A key mechanism of this process is the reduced expression of vascular endothelial (VE) cadherin (CD144) in the ECs contact area [340,341].

The fourth and final step is the remodeling of the actin cytoskeleton of ECs and migrating cells, which is a prerequisite for transendothelial leukocyte migration [341].

All the processes of exudative reaction and leukocyte migration to the inflammation focus are mainly observed in postcapillary venules, which are lined by type 2 ECs [59]. In contrast, the arterial part of the vascular network is not characterized by these processes, although in atherosclerosis, there may be a significant increase in the transendothelial migration of leukocytes in arteries [342,343]. For instance, treatment with oxLDL at a concentration of 40 µg/mL increased the expression of E- and P-selectins, VCAM-1, and ICAM-1 in a culture of human coronary artery ECs (HCAEC) mediated through LOX-1 [344]. In patients with diabetes, elevated levels of soluble VE-cadherin (sCD144) in plasma were identified as a significant risk factor for coronary heart disease compared to all other traditional risk factors [345,346,347].

In the elderly, problematic areas of the arteries for atherosclerosis are characterized by the presence of abnormal multinucleated ECs [347]. In this case, the basal endothelial membrane thickens, increasing its affinity to LDL, especially for modified LDL. This condition may result in the disruption of endothelial barrier function, either by the separation of ECs from the basal membrane or by the opening of dense contacts between ECs [348].

Chemokines are a group of small cytokines that are known for their ability to regulate leukocyte recruitment to sites of inflammation by inducing chemotaxis. This is achieved by binding chemokines to their receptors on white blood cells, leading to changes in cellular behavior, such as the rearrangement of actin and cell shape, allowing for leukocyte migration. In particular, the migration of neutrophils and monocytes across the endothelial barrier is regulated by CXCL8 (IL-8) and CXCL1, which can be secreted directly by ECs [349,350]. CXCL1, in particular, enhances the adhesive and pro-inflammatory properties of ECs in an autocrine manner, exerting an atherogenic effect. However, CXCL1 expression can be inhibited by miR-181b, which inhibits NF-κB in the atherosclerotic endothelium, exhibiting anti-atherogenic properties [351].

CXCL12 (SDF-1) is secreted by stromal cells [352]. It interacts with glycosaminoglycans in the endothelium, which are essential for its stability and presentation to leukocytes. CXCL12 acts through the CXCR4 receptor, which is widely expressed in various cell types associated with cardiovascular disease, including ECs, VSMCs, T cells, B cells, neutrophils, monocytes, and Mf [352]. While CXCL12 promotes the migration of these cells, it also accelerates the healing process of damaged vessels and supports ischemic neovascularization [353]. However, CXCL12 is considered to be a proatherogenic factor due to its involvement in dyslipidemia, angiogenesis, plaque destabilization, thrombus formation, neointimal hyperplasia, and pro-inflammatory effects against vascular endothelium [354]. Moreover, an increase in LDL levels is associated with the activation of CXCL12 expression, which is inversely proportional to HDL levels [355].

CXCL9, CXCL10 (IP-10), and CXCL11 are secreted by monocytes, ECs, and fibroblasts. In response to IFN-γ and TNF-α, they act through the CXCR3 receptors and promote the migration of monocytes, T lymphocytes, dendritic cells (DCs), and natural killer (NK) cells [356]. In patients with morbid obesity and metabolic syndrome, CXCL10 and CXCL11 are associated with insulin resistance and increased leukocyte adhesion on the endothelium [357]. The chemokine CCL20 (acting via the CCR6 receptor) is important for monocyte recruitment in atherosclerosis [358]. On the other hand, CCL5 (RANTES) is secreted by activated platelets and ECs and interacts with several receptors (CCR1, CCR3, CCR5), mediating monocyte/macrophage infiltration in atherosclerotic lesions [359].

The literature supports the importance of the CCL2 (MCP-1)—CCR2 axis in human atherosclerosis, highlighting its relevance and therapeutic potential [360]. CCL2, primarily secreted by monocytes and Mf, is bound to the endothelial glycocalyx and is considered the strongest chemoattractant for monocytes. Endotheliosis involves multiple interrelated processes that contribute to the development of arterial wall inflammation in atherosclerosis. In addition to ECs, other factors play a role in atherosclerotic inflammation.

### 4.3. Mast Cells in Atherogenesis and Aging

Adventitial mast cells (MCs) are located near the vasa vasorum of the aortic and coronary artery walls and in the intima, close to the lumen endothelium, as it lacks microvessels and is absent in normal conditions. MCs are powerful participants in the inflammatory response in various tissues, including both intimal and adventitial layers of atherosclerotic arteries [361], and contribute to the onset and progression of atherosclerosis [362]. Thus, activated MCs release histamine, heparin, neutral proteases (tryptase and chymase), ROS, and cytokines such as TNF-α, IL-6, CCL2, and CCL5 at the site of atherogenesis [361,363,364]. Due to their ability to release angiogenic compounds, histamine and proteases, MCs not only cause microvessel growth into the intima but also rupture new vessels, leading to intramembranous hemorrhage [365]. Immune complexes (including IgG-oxLDL), DAMPs, complement anaphylaxins (C3a and C5a), and pro-inflammatory cytokines act as MCs activators [362,366,367], while SR-B1 is involved in TC degranulation [367].

Several studies have shown a relationship between MCs and aging. In particular, an increase in the number and activity of MCs in the mesentery [368], heart, and kidneys were observed in aging rats [369]. In aging, MCs may also be involved in the pathogenesis of osteoporosis [370]. Aging causes basal activation of lymphoid MCs, which, in turn, limits the beneficial function of immune cells [371]. Taken together, these changes play an important role in the pathogenesis of inflammation and immunity associated with aging.

In humans, serum levels of tryptase, a marker of MCs activation and degranulation, have been found to be increased in aging, particularly in obesity [372,373]. However, some studies have reported a decrease in degranulation and an increase in MC numbers with aging [374]. In addition, activated MCs may contribute to the pathogenesis of neurodegenerative diseases in aging by disrupting the permeability of the blood-brain barrier, thereby allowing toxins and immune cells to enter the brain [375]. Overall, MCs are systemically involved in the formation of allostasis in the aging organism and thus participate in the pathogenesis of many aging-related chronic diseases, including atherosclerosis. In classical inflammation, particularly in immediate-type allergies, the role of mast cells is primarily to ensure local exudative and vascular reactions. However, in aging and low-grade inflammation, MCs’ function is less demonstrable and is associated with remodeling the microvascular network and modulating the function of different types of immune cells. At the same time, pathologically activated MCs are also capable of maladaptive effects in the tissues where they are localized.

### 4.4. Role of Platelets in the Pathogenesis of Atherogenesis

Platelets play a crucial role in hemostasis, preventing blood loss from injured blood vessels. Evolutionarily, platelets are thought to have originated from blood phagocytes, and their nuclear-free forms first appeared in mammals [59]. As a result, platelets share many functional similarities with phagocytic leukocytes, including the expression of receptors for TLR, SR, various complement factors, Fc receptors, and CD154 receptor (CD40 ligand on antigen-presenting cells). Platelets also express adhesion receptors such as P-selectin and integrins and are able to produce NO via eNOS and iNOS. Upon activation, platelets release various factors, including cationic proteins, proteases, cytokines (such as IL-1β, IL-6, IL-8, TNF-α, and various chemokines), and clotting factors [376,377,378,379,380,381,382,383,384,385,386].

Thus, platelets serve two primary functions: (1) participating in hemostasis to stop bleeding in case of vascular damage and (2) contributing to the development of various types of inflammation. In the latter case, the role of platelets and microthrombi in postcapillaries in the development of an exudative-vascular reaction in the focus of canonical inflammation is evident. At the system level, this process is one of the characteristic phenomena of systemic hyperinflammation and disseminated intravascular coagulation (DIC) [60].

However, platelets can also be involved in low-grade inflammation, specifically in the formation of moderate thrombophilia at the systemic level (not reaching the criteria for DIC) and, at the local level, in increased endotheliosis, endothelial damage, and leukocyte migration in arterial regions prone to atherosclerosis development [387].

The proatherogenic effects of activated platelets are related not only to their secretory function but also to their direct adhesion to EC, extracellular matrix structures, and their aggregation with monocytes and other leukocytes. Platelets can be activated by atherogenic lipoproteins, DAMP and PAMP, immune complexes, active hemostasis factors, some miRNAs, adhesive interactions, and complement [388,389,390,391].

In the presence of an intact and healthy endothelium and laminar blood flow, circulating platelets remain in a resting discoid state near the apical surface of ECs, without forming adhesive contacts. This is due to the antiadhesive properties of normal endothelium. After tissue injury, platelets are exposed to thrombogenic molecules present in the subendothelial tissue, such as collagen, von Willebrand factor (VWF), laminin, and thrombospondin, leading to their activation and adhesion to the damaged endothelium. Even with an abrupt change in blood flow shear stress or a change in the vascular microenvironment, circulating platelets can quickly perceive these signals and react with a subsequent interaction with activated ECs [389].

The formation of platelet-macrophage aggregates shifts Mf to a more pronounced pro-inflammatory phenotype [392], and in patients with atherosclerotic diseases, platelet activity correlates with Mf production of inflammatory cytokines [393]. Platelets express many surface receptors that interact directly (through receptors) or indirectly (through molecular adhesion cofactors) with collagen and other extracellular matrix structures of the basal endothelial membrane and atherosclerotic plaque [394,395]. Among these receptors, the integrin α2β1 (GPIa/IIa) and Ig-like receptor glycoprotein VI (GPVI) are the most important. Integrin α2β1 predominantly mediates adhesion, while GPVI is a strong platelet activator after contact with collagen. The presence of SR-B2 (CD36) on the surface of platelets has been shown to promote their hyperactivity upon contact with oxLDL [396].

Furthermore, platelets have been shown to play a role in modulating the differentiation of monocytes into Mf and influencing the ability of Mf to accumulate lipids and become foam cells [397]. When activated, platelets secrete numerous chemokines, including CXCL4 (PF4), CCL5 (RANTES), CXCL12 (SDF-1α), CXCL16, and others, which initiate or promote local inflammatory processes at sites of vascular injury. These processes are mainly mediated by the recruitment of circulating hematopoietic stem cells, neutrophils, monocytes, and lymphocytes to the vascular wall [398].

One of the key factors contributing to vascular complications in patients with type 2 diabetes mellitus (DM2) is increasing atherosclerosis. Hyperglycemia and insulin resistance can disrupt normal endothelial and platelet function, leading to the development of a prothrombotic state in DM2 patients [399]. These pro-atherogenic effects increase the risk of vascular complications in DM2 patients.

In summary, platelets act as inflammatory factors and may be involved in the pathogenesis of atherosclerosis, from the initial atherogenic manifestations of endotheliosis to their involvement in the inflammatory microenvironment of the atherosclerotic plaque.

### 4.5. Role of Lipoprotein Metabolism Dysfunction in the Pathogenesis of Atherosclerosis

The accumulation of plasma cholesterol and atherogenic lipoproteins (Lp), especially modified LDL, is a crucial metabolic condition for the development of atherosclerosis. All Lp have a hydrophobic lipid core surrounded by a hydrophilic envelope consisting of phospholipids and apolipoproteins (apo). Lp(a) has the highest atherogenic activity among native Lp and differs little from LDL in the composition of the lipid core and protein envelope [400,401]. However, Lp(a) contains not only the aroB-100 protein but also the apo (a) protein, which is structurally similar to plasminogen. As a result, Lp(a) has a high affinity for the extracellular matrix and rapidly accumulates in the vascular wall. Apo(a) undergoes oxidation and partial proteolysis by metalloproteinases, which contribute to the development of an atheromatous plaque. Recent review publications, such as those by Mehta (2022) [402], Lu (2022) [403], Di Giovanni (2023) [404], Ronda (2023) [405], Bellot (2023) [406], Guo (2023) [407], Močnik (2023) [408], Kim (2023) [409], and Poznyak (2023) [410], have detailed the pathogenetic role of impaired Lp metabolism in atherogenesis. It has been shown that all Lp-containing apoB-100 is associated with atherosclerosis. On the other hand, HDL helps remove cholesterol from Mf and VSMCs, thereby inhibiting the formation of foam cells. As mentioned in Section 4.2.3, effective arterial lymphatic drainage is important to allow HDL backtransport to the liver, where excess cholesterol undergoes oxidation and conversion to bile acids.

Initially, neutral fat (triacylglycerol) and cholesterol esters are synthesized in the liver, mainly from incoming blood glucose and FFA. These lipids then enter the bloodstream as very low-density lipoproteins (VLDL). Under the action of lipoprotein lipase, VLDL loses most of its neutral fat and becomes LDL, which transports cholesterol and esters to various cells by binding, via apolipoprotein B-100 (apoB-100), to the cellular receptor for LDL (LDLR).

ApoB-100 is the primary structural component of atherogenic lipoproteins, including VLDL, LDL, and Lp(a), while apoB-48 is specific to chylomicrons that are formed in the intestine to carry exogenous lipids to the liver and peripheral tissues. ApoA-I and apoA-II provide the framework for the lipidation of HDL, while apoC-II, apoC-III, and apoE are involved in the metabolism of triacylglycerol-rich lipoproteins. Apo(a) covalently binds to apoB-100 to form Lp(a). However, apoB-100 and apoB-48 are insoluble apolipoproteins and cannot remain in the blood in an Lp-free state. In contrast, other apolipoproteins are soluble and can exist as free proteins in plasma and be freely exchanged between lipoproteins.

After transendothelial transfer, amino acids of apoB-100 in LDL bind to proteoglycans of the intima, leading to LDL retention in the arterial wall. Some LDL undergoes multiple modifications, including oxidation, and then binds to SRs on Mf, leukocytes, ECs, and VSMCs, causing local inflammatory and immune responses, as well as the formation of foam cells. Foam cells mainly originate from Mf and transdifferentiated VSMCs and enhance the inflammatory cascade, promoting further migration into the intima, proliferation, and transdifferentiation of VSMCs, ultimately leading to the formation of atherosclerotic plaques. At the same time, VSMCs lose contractile function and acquire a pro-inflammatory secretory phenotype.

The accumulation of Lp in the blood is promoted by several factors, including excessive intake of glucose and FFA from the blood into the liver, impaired formation of phospholipids in the liver leading to dyslipoproteinemia, inadequate reverse HDL cholesterol transport, oxidative stress, hyperglycemia, and other modifications of atherogenic lipoproteins, as well as genetic features and defects in the formation of individual forms of apolipoproteins and lipoprotein cell receptors.

### 4.6. Role of Metabolic Dysfunction in the Pathogenesis of Atherosclerosis

Low-grade inflammation is a common factor in the pathogenesis of morbid obesity, metabolic syndrome, and DM2. This inflammation is directly linked to metabolic damage factors.

Several metabolites, including glucose, FFA, and many amino acids, have regulatory functions by acting on cellular receptors, allosteric centers of key enzymes in metabolic pathways, and various links of intracellular signaling pathways [411,412,413,414,415]. These metabolites are direct participants in forming network information-regulatory fields at the cell, tissue, and holistic levels of organisms. In pathology, these fields are distorted due to the accumulation of dysfunctional (dysregulatory) systems with the pathological character of forward and backward regulation.

At the organism level, these effects are largely associated with metabolic cycles, including the glucose fatty-acid cycle (Randle cycle) and glucose-alanine cycle. These cycles integrate metabolic processes in major facultatively glycolyzing organs, such as the liver, adipose tissue, and skeletal muscle, to regulate blood concentrations of the main energy metabolites, primarily glucose and FFA.

However, the functions of these cycles are not characterized by mobility and are complemented by neuroendocrine regulation. For example, the main insulin-dependent transmembrane glucose transporter, known as GLUT4, is mainly localized in adipocytes and muscle tissue [416,417]. On the other hand, the main glucose transporter in hepatocytes (GLUT2) functions depending on the ratio of intra- and extracellular glucose and the action of several factors on hepatocytes, including FFA and pro-inflammatory cytokines, that contribute to insulin resistance [418,419].

In individuals with obesity, there is a decrease in the sensitivity of GLUT4 to insulin. This leads to an excessive accumulation of FFA and glucose in the blood. The activation of gluconeogenesis in the liver and the dysregulatory effects of excess FFA contribute to this effect. In addition, pro-inflammatory tissue stress in the liver (hepatosis), which can be complicated by nonalcoholic fatty liver disease, exacerbates the process of forced glucose synthesis and dysfunction of lipoprotein formation in the liver [420,421,422]. The accumulation of lipotoxicity factors and pathological glycation products in the blood can act as inducers of chronic low-grade inflammation in the endothelium and other tissues [423].

Furthermore, lipolysis in adipocytes and increased FFA levels in the blood are correlated with hepatic Mf activity in patients with non-alcoholic fatty liver disease [424]. Recent studies have also shown a positive correlation between plasma levels of hydrophobic branched-chain amino acids, such as leucine, valine, and isoleucine, and the occurrence of atherogenic metabolic disorders [425].

On the other hand, an imbalance in histohormone production in the facultatively glycolyzing organs can also contribute to the development of low-grade inflammation in obesity. For instance, the development of low-grade inflammation in adipose tissue causes an increase in the number and pro-inflammatory activity of stromal Mf there [426,427,428]; aging intensifies this process [429]. The production of pro-inflammatory cytokines by Mf in adipose tissue can have not only local but also systemic effects, including increased gluconeogenesis and insulin resistance in the liver [430].

At the same time, an imbalance in lipokine production in adipocytes develops with a predominance of pro-inflammatory, atherogenic factors (leptin, chemerin, resistin, IL-6, and others) over anti-atherogenic factors (FGF-21, CTRP9, omentin, progranulin, adiponectin, and others) [431,432]. These dysfunctions contribute to endothelial dysfunction, hypertension, and the formation of pro-inflammatory allostasis in general.

Therefore, low-grade inflammation is associated with metabolic dysfunction and insulin resistance at the level of individual organs and the body as a whole. This variant of the inflammatory process can be considered a typical complication of aging, providing an additional stimulus for the development of atherosclerosis and other cardiovascular diseases.

### 4.7. Summary

Atherosclerosis is a complex pathology that involves metabolic, immune, and inflammatory disorders at both local and systemic levels. Endotheliosis and local low-grade inflammation within the arterial wall are closely linked to the development of atherosclerosis. Systemic low-grade inflammation and associated pathologies such as morbid obesity, metabolic syndrome, and DM2 are consequences of increased pro-inflammatory systemic tissue stress that accompanies aging, known as “inflamm-aging.” Low-grade inflammation is the underlying cause of negative dynamics in many pathologies, including atherosclerosis. Consequently, vascular complications arise due to the formation of unstable atherosclerotic plaques, which are directly related to increased local inflammation and the involvement of various immune mechanisms.

## 5. Immune Mechanisms of Atherosclerosis Pathogenesis

### 5.1. Macrophages

The immune system in the arteries в нoрме primarily consists of stromal Mf situated in the intima and adventitia. These Mf are an essential component of innate immunity and play a pivotal role in inflammation, wound healing, and defense against infection and other harmful factors. However, stromal Mf not only serve as immunocytes but also perform various specialized functions to maintain tissue homeostasis. As a result, stromal Mf acquire a tissue-specific phenotype, such as osteoclasts, chondroclasts, microglia, mesangial Mf, Kupffer cells (also known as stellate Mf of liver sinus capillaries), CD169+ Mf of secondary lymphoid organs, and more [433,434].

The majority of stromal Mf are of embryonic origin and maintain their numbers in tissues through proliferation [433,434]. However, in some organs, embryonic Mf are replaced by other Mf originating from bone marrow monocytes. This replacement primarily occurs in mucous membranes and lymphoid organs, as well as in the heart and arteries [433,434,435]. Mf of monocytic origin tend to have greater immune and pro-inflammatory activity, while embryonic cells exhibit more homeostatic activity. The number of stromal Mf of bone marrow origin increases with age, including in arteries and adipose tissue [436,437].

Under conditions of intimal lipidization, stromal Mf attempt to excrete excess lipids, but this process is quickly overwhelmed, turning Mf into “foam cells” [438]. Simultaneously, upon activation, similar to that occurring in hypercholesterolemia or at sites of impaired blood flow, monocytes are recruited into the subendothelial space and then differentiate into pro-inflammatory Mf, capable of forming foam cells [439]. This process is enhanced when Mf release inflammatory cytokines, primarily TNF-α and IL-1β, causing the expression of adhesion molecules and chemokine production in ECs [439].

At the same time, Mf themselves produce chemokines that attract monocytes and other leukocytes (from the bloodstream) and VSMCs (from the media) to the focus of atherogenesis. Mf can also be released from the plaque through interaction with chemokines CCR7, CCL19, and CCL21, which are produced by ECs (CCR7) and arterial stromal cells (CCL19, CCL21) [440].

The process of foam cell accumulation in the intima is realized already at the stage of lipoidosis during the formation of lipid spots and bands. With that, the main effectors of atherogenesis become inflammatory Mf of monocytic origin, which are replenished by monocyte migration and then by the proliferation of recruited Mf. At the stage of atherosclerotic plaque formation, transdifferentiated, modified VSMCs, primarily with a macrophage-like phenotype, are also actively involved in the formation of foamy cells [441]. During plaque formation and local hypoxia, vessels ingrown into the intima from the media, through which monocytes and other leukocytes can migrate [442]. The transport function of the vasa vasorum also increases with the increase in plaque size and accumulation of cells in it.

When cholesterol efflux is activated by HDL function, plaque Mf acquire a pro-resolving M2-like phenotype, releasing anti-inflammatory cytokines such as IL-10 and TGF-β, thereby promoting tissue repair or fibrosis [443]. In addition to M2 Mf, fibroblast-like VSMCs producing collagen and other extracellular matrix proteins contribute to fibrosis and stable plaque formation [444,445].

Conversely, the dominance of M1 Mf, macrophage-like, and osteoblast-like VSMCs is a prerequisite for unstable plaque formation [444,445]. M1 Mf are predominantly localized along the perimeter of the lipid-protein core and in the cap of the unstable plaque, while M2 Mf and myofibroblasts localize in the area of fibrous surroundings and intimal vascularization. M1 Mf not only inhibit the structural function of M2 but also actively secrete matrix metalloproteinases (MMP-1, MMP-3, MMP-10, MMP-12, MMP-4, and MMP-25) that degrade the fibrous membrane and contribute to tissue destruction zones from destroyed extracellular matrix structures, cellular necrosis, and unfinished apoptosis [442,446].

M1 cells are characterized by a pronounced pro-inflammatory phenotype [439,442,447,448], which includes:High expression of pro-inflammatory cytokines TNF-α, IL-1β, and IL-12;iNOS expression;High expression of TLRs, other PRRs, and SR-B2 (CD36), which is the most functionally related to TLRs of all SRs;High level of transcription factor (TF) expression: HIF-1, NF-kB, STAT.M1 cells are more active in secreting chemokines, such as CCL2, CCL3, and CCL5 than M2 cells;In some cases, M1 cells may actively form NLRP3 inflammasomes, which are associated with pro-inflammatory cellular stress, hyperproduction of IL-1 and IL-18, and pyroptosis.

The dual function of SR-B2 (CD36) on ECs, platelets, and various immunocytes, particularly M1 cells, should be considered. SR-B2 (CD36) acts as (1) a signal transducer from PAMPs, DAMPs, and oxLDL and (2) an FFA transporter. However, when FFA β-oxidation and aerobic phosphorylation function are deficient, it can increase lipotoxicity and contribute to mitochondrial dysfunction [449].

In contrast, M2 cells are characterized by a more pronounced expression of SRs in general, especially those involved in efferocytosis (Table 1) and the mannose receptor, SR-E3 (CD206), as well as arginase1, which is involved in the synthesis of extracellular matrix proteins. M2 cells also express TFs, such as STAT3, STAT6, PPAR, and anti-inflammatory cytokines (IL-10, TGF-β) [441,444,450]. M2 cells utilize the aerobic breakdown of glucose and FFA to meet their energy requirements, while M1 cells use glycolysis [448,450,451]. This difference is related to several factors, including the fact that M1 cells are typically located in a more hypoxic environment than M2 cells. Additionally, the absence of aerobic processes in M1 cells reduces the cell’s dependence on mitochondrial and oxidative stress, which are more pronounced in M1 cells. With glycolysis, there is no competition for oxygen between oxidative phosphorylation and ROS formation, which can be formed not only in mitochondria but also during microsomal oxidation. Moreover, M1 cells release the end product of glycolysis (lactate), further acidifying the extracellular environment, thereby enhancing local pro-inflammatory mechanisms.

In the bloodstream, at least two subpopulations of monocytes are distinguished, with the most significant of them (up to 90%) able to differentiate into M1, while the other can differentiate into M2 [438,441]. However, under the influence of cytokines and other environmental factors, a morphofunctional drift of Mf in the M1-M2 range is possible [75]. Thus, the division into classical (M1) and alternative (M2) polarization of Mf is, to some extent, conditional [436].

In vitro, the current classification of Mf formed from monocytes includes at least 10 distinct subpopulations [452]. This differentiation is probably even more complex in vivo [453,454,455,456]. Nevertheless, the various variants of Mf differentiation can be simplified into four conditional types: M1, M2a, M2b, and M2c, including atherosclerosis [436]. It is also possible to distinguish special variants of Mf differentiation, such as M4, as well as hemoglobin-stimulated Mf-M(Hb), which are also designated as Mhem or HA-Mac [448,456,457].

M(Hb) are characterized by antioxidant and anti-atherogenic activity and are located in the focus of atheromatosis near the vasa vasorum and associated with intrabladder hemorrhages [458,459]. These cells highly express SR-I1 (CD163) and a heme-dependent transcription factor (ATF1) that induces the expression of hemoxygenase 1 and LXR-β (liver X receptor). M(Hb) are also involved in hemoglobin clearance through erythrocyte phagocytosis and increases cholesterol efflux by expression of LXR-β-dependent genes [458,459]. In turn, M4 expresses genes responsible for the induction of extracellular matrix-destroying proteases such as MMP7, MMP8, and MMP12 [de Sousa, 2019] and genes facilitating foam cell formation [455].

Each Mf type cooperates with complementary CD4 T-helper (Th) types and other cells of the immune system during the development of immune inflammation, thereby forming certain immune response vectors (I) in the focus of inflammation: I1 (M1-Th1); I2 (M2a-Th2); I3 (M2b-Th17); I-reg (M2c-Treg) [75].

Regarding atherosclerosis, on the one hand, oxLDL uptake by Mf can be considered a protective mechanism since they remove cytotoxic elements from the intima. On the other hand, the increased migration of monocytes into the intima and their subsequent differentiation into inflammatory Mf and then foamy cells leads to the formation of an atherosclerotic plaque, which can become unstable and lead to severe hemodynamic disorders.

### 5.2. Cellular Immune Response Vectors in Atheromatous Inflammation

The process of atheromatosis during the formation of stable and unstable plaques resembles chronic classical inflammation with a fibrotic component. This type of inflammation involves complex intercellular interactions, particularly between Mf and T-helper cells. These interactions can be categorized into four vectors of immune response (I1, I2, I3, I-reg) that are active in the site of inflammation (see Table 2).

It is important to consider that Th differentiation, as well as Mf, is characterized by plasticity. This means that certain spectrums of cytokines can cause transformations, such as Treg into Th17 or Th2, Th17 into Th1, or even the formation of functionally unstable T-cells with a mixed phenotype. For example, Th2 cells can transform into CD4^+^ T-cells that can simultaneously produce cytokines of competitive Th (Th2 and Th1), specifically IL-4 and IFN-γ [471,472,473]. Other cells with an intermediate Th1/Th17 phenotype simultaneously express TFs, such as T-bet and ROR, which allows for the simultaneous production of functionally different types of cytokines (IFN-γ, GM-CSF, and IL-17) [474]. Additionally, Th1, Th17, and Th2 can be divided into more specific subpopulations [475,476].

Furthermore, mature Treg cells can undergo epigenetic modifications, which can cause them to lose FOXP3 expression and transform from anti-inflammatory to pro-inflammatory cells [477].

Another group of cells to consider are γδ T cells, which sit at the border of adaptive and innate immunity and are also functionally plastic [478,479]. Studies have shown that γδ T cells in mice are predominantly found in the aortic root and contribute to the early formation of atherosclerotic lesions, plaque necrosis, and inflammation at this site [480]. However, in other studies, the immunological activity of total γδ T-cells in the blood of patients with coronary heart disease was significantly lower than that of healthy individuals [481]. The exact role of this subgroup of T cells in atherosclerosis is still unclear [482].

In general, epigenetic analysis has shown that the epigenome of cells involved in atherogenesis, such as ECs, fibroblasts, VSMCs, and various immunocytes, exhibits plasticity and heterogeneity [483]. The classification of pro-inflammatory immune responses (I1, I2, I3, I-reg) is conditional, and morphofunctional changes of immunocompetent cells occur within certain corridors. The boundaries and integration of these corridors are influenced by immune response triggers, immunocyte maturity, genetic and epigenetic interactions, inflammation activation factors, cytokine network features, and extracellular communication mechanisms like intercellular transvesicular exchange [484,485]. Additionally, even competing immune responses such as I1 and I2 have zones of overlap and functional cooperation, and the interaction between M1 and M2 Mf can lead to progressive interstitial fibrosis [486]. Overall, I1 factors promote atherosclerosis and unstable plaque formation by producing pro-inflammatory cytokines and chemokines. In contrast, M2c and Treg macrophages suppress inflammation and reduce plaque size, while M2a and Th2 promote fibrosis and plaque stability [436,439,440,442,447,448,487].

M2b Mf, like M1, produce pro-inflammatory cytokines such as IL-1, IL-6, TNF-α, and others, contributing to plaque destabilization [442]. Th17 cells not only activate M2b but also attract neutrophils to the focus of atherogenesis, which increases tissue destruction [171,488] (Figure 4). IL-17A signaling activates various downstream pathways, including the NF-κB pathway, intensifying inflammation and atherosclerosis in ECs, M2b Mf, and VSMCs [489,490]. The metabolic profiles of Th17 cells and Th1/M1 cells are similar, including the functions of glycolysis and hypoxia in the formation of these areas of the immune response [491,492]. However, some studies have also shown an anti-atherogenic role of IL-17 [428,493]. In addition to pro-inflammatory cytokines, M2b Mf also secrete significant amounts of IL-10, which is an IL-1 inhibitor and antiatherogenic factor [428,493]. The function of M2b Mf in unstable plaque formation is ambiguous, especially in I-reg deficiency.

B-cells may also be involved in the focus of atherogenesis and interact there with Mf and other lymphocytes. B-1 lymphocytes, which are on the edge of innate and adaptive immunity, exhibit their anti-atherogenic properties by secreting natural IgM antibodies that contribute to the removal of oxLDL from the bloodstream. However, antigen-specific B-2 cells stimulate Th1 cells and DCs to play a proatherogenic role [493].

### 5.3. Antigen-Specific Mechanisms of Atherosclerosis

The involvement of T cells, B cells, and DCs in atherogenic inflammation, as well as the presentation of antigenic oligopeptides to CD8 and CD4 T-lymphocytes via major histocompatibility complex (MHC, HLA) proteins, provides evidence for the presence of antigen-dependent mechanisms of atherosclerosis [495,496,497]. Upon encountering antigens, naive T cells transform into effector memory T cells, particularly Th1 cells that produce interferon-gamma and proatherogenic cytokines [497]. The accumulation of monoclonal and oligoclonal effector CD4 T cells and cytotoxic CD8 T cells, including those recognizing autoantigens such as oxLDL, in atheroma contributes to the formation of unstable plaques [498].

Mature, antigen-specific T cells interact with Mf expressing HLA-DR and cofactor receptors CD40 and B7 (CD80/CD86), which are necessary for their antinational interaction with T cells [499]. HLA-DRs are also expressed in atherosclerotic plaques and on VSMCs, which is not typical of normal arteries. Moreover, the involvement of B-2 lymphocytes in the pathogenesis of atherosclerosis indicates the participation of antigen-specific immunoglobulin (Ig) and follicular helper T cells (Tfh) in this process [500].

In advanced stages of atheromatosis development, T- and B-zone cell proliferations are present in arteries with active germinal centers, including DC and plasmocytes, while the formation of lymphatic follicles in the adventitia of atherosclerosis-affected arteries is designated as arterial tertiary lymphoid organs (ATLO) [501]. Analysis of ATLO immune cell subpopulations indicates antigen-specific T- and B-cell immune responses in the adventitia of atherosclerotic arteries. Additionally, ATLOs contain innate immune cells, including a large component of inflammatory Mf, B-1 cells, and an aberrant set of antigen-presenting cells, primarily DC. ATLOs are also characterized by neoangiogenesis, lymphangiogenesis, high endothelial venules (type 2 ECs), naive lymphocyte recruitment, and central and effector memory CD4 T cells [501,502].

Infectious antigens may contribute to the development of atherosclerosis, as seen in periodontal diseases [503]. Mayr et al. (2020) have shown a high reactivity of T cells against chlamydial antigens in atherosclerotic plaques [504]. Chronic viral infections [505] and COVID-19 [506] are also suggested to play a role in the development of cell aging and atherosclerosis in the literature. However, atherosclerosis is not generally considered an infectious disease, allowing us to focus on the autoimmune mechanisms of this pathology, which are primarily associated with I1 and I3 in atheroma (Table 2).

Multiple studies have demonstrated that several systemic autoimmune diseases are associated with accelerated development of atherosclerosis. Patients with antiphospholipid syndrome have been found to have high levels of oxLDL-binding antibodies detected in autoimmune-mediated atherothrombosis [507]. Additionally, autoantibodies to oxLDL are found in atherosclerosis, and these cross-react with antiphospholipid antibodies [508]. Furthermore, immunopathological problems such as systemic lupus erythematosus, antiphospholipid syndrome, and rheumatoid arthritis have been shown to accelerate atherogenesis [509,510,511,512,513].

In the context of atherogenesis, oxLDL interacts with CRP to form oxLDL-CRP complexes, which maintain not only vascular inflammation but also trigger autoimmune reactions [514]. Patients with atherosclerosis have significantly elevated autoantibodies against HSPs, and T-lymphocytes specifically react to HSPs as autoantigens in atherosclerotic plaques [515]. Additionally, IgG and IgM antibodies to oxLDL are present in animal and human plasma, and they form immune complexes with oxLDL in atherosclerotic lesions [515]. Furthermore, IgG antibodies to oxLDL and IgG antibodies against apoB-100 are positively associated with the presence of cardiovascular disease [516]. A proatherogenic role of IgG and IgG immune complexes has been revealed through the activation of Mf in atheromas via Fc receptors [517]. However, the specific influence of antibodies on atherosclerosis remains to be elucidated compared to other B-cell functions, such as cytokine production or the influence of B cells as antigen-presenting cells [518].

In summary, while infectious antigens can contribute to the development of atherosclerosis, autoimmune mechanisms are also important in this pathology. Multiple systemic autoimmune diseases are associated with accelerated atherogenesis, with autoantibodies against oxLDL and HSPs and IgG and IgG immune complexes playing a proatherogenic role. The specific role of antibodies in atherosclerosis compared to other B-cell functions requires further investigation.

Meanwhile, Fab-fragments of IgG or single-stranded variable fragments of antibodies against oxLDL can reduce the formation of foam cells [519], indicating that the neutralizing functions of antibodies against oxLDL may play an important protective role in atherosclerosis. Moreover, IgG against oxLDL can neutralize and facilitate the clearance of oxLDL by forming oxLDL-IgG complexes that are rapidly eliminated by Fc-receptor Mf in the liver and spleen [520]. Adoptive transfer of human IgG1 against modified apoB-100 can reduce the degree of atherosclerosis and autoimmune oxLDL epitopes in the blood of recipient mice [521]. Therefore, certain types of antibodies against modified LDL may have potential therapeutic value through the use of passive or active immunization [518]. However, further large-scale studies are required to definitively establish the correlation between different oxLDL-binding antibody isotypes and cardiovascular disease [522].

Additionally, it is necessary to consider the possibility of the formation of autoantigens and haptens (primarily phospholipids) not only as a result of the molecular modification of atherogenic lipoproteins but also the potential formation of other autoantigens due to a variety of reasons [496,523,524,525,526]. These reasons include molecular mimicry of microbial proteins during atheroma infection, “bystander activation”—the release of autoantigens from tissue damaged by inflammation, polyclonal activation of lymphocytes, “epitope spreading”—where the targets of autoimmune responses are expanded by including other epitopes on the same protein or in other proteins in the same tissue, systemic or local immune dysfunction, Treg deficiency, and increased mutagenesis (for example, as a result of oxidative cellular stress) and decreased efficiency of utilizing abnormal proteins. Another potential cause of immune dysfunction in atherosclerosis is increased stem cell proliferation, which can accelerate negative somatic evolution and proliferation of myeloid cell clones with driver mutations [527].

Thus, there is currently no doubt about the involvement of autoimmune mechanisms in the development of atherosclerosis, both as inducers of inflammation and as factors that facilitate the clearance of modified Lp from the bloodstream. Simultaneously, there is no convincing reason to deny the key role of scavenger receptor (SR)-dependent uptake of modified LDL by Mf and VSMCs in atherogenesis, which subsequently leads to foam cell formation.

### 5.4. Summary

In summary, the development of atherosclerosis is mediated by a complex interplay between innate and adaptive immune mechanisms, with inflammation primarily driven by inflammatory Mf and foam cells. Modified Lp and cholesterol deposition in the intercellular matrix and foam cells serve as major triggers of inflammation, with autoimmune and infectious factors also playing a role. The different immune reactivity vectors (I1, I2, I3, I-reg) are involved in the inflammatory process and may be localized in different layers of the atheroma, with the ratio of these immune responses affecting the severity of inflammation and stability of the atherosclerotic plaque. Despite the involvement of autoimmune mechanisms, the role of SR-dependent uptake of modified LDL by Mf and VSMCs with the subsequent formation of foam cells cannot be ignored in the development of atherosclerosis.

## 6. General Evaluation of the Inflammatory Process in Atherosclerosis

The development of atherosclerosis involves a local low-grade inflammation during the pre-atherosclerosis stage, characterized by manifestations of endotheliosis. In the initial subclinical stage of atherosclerosis, there is the formation of lipid spots and bands in the arterial intima (lipoidosis). This stage is associated with the appearance of frothy cells, namely accumulated cholesterol Mf and, in some cases, VSMCs, and deposition of lipids in the intercellular matrix of the intima. It is noteworthy that lipid spots and bands can be detected even in childhood, suggesting that lipoidosis can develop under conventionally physiological conditions without overt endotheliosis manifestations.

Lipoidosis can be considered a transitional stage to the development of the next stage of atherosclerosis, which is productive inflammation in the formation of atherosclerotic plaque (atheromatosis). The commonality of the molecular mechanisms of CS in atherosclerosis and low-grade inflammation suggests the proximity of epigenetic cellular changes in atherogenesis to other pro-inflammatory processes of the elderly, such as osteoporosis [528] and neurodegenerative diseases [529].

Abnormal states of autophagy, pyroptosis, and ferroptosis in vascular cells, including ECs, VSMCs, and Mf, associated with oxidative stress, pro-inflammatory cellular phenotype, and other manifestations of CS, play a key role in the development of atheromatosis [114]. The formation of an atherosclerotic plaque involves the destruction of the internal elastic lamina, large-scale migration of VSMCs into the focus of atherogenesis, and local vascularization of the media side intima. In addition, not only the media but also the adventitia is involved in the process, where tertiary lymphoid organs are formed. The peripheral nervous system uses the adventitia to regulate atherogenesis processes through the existence of neuroimmune interfaces in the arteries [282,530]. In these cases, the interactions between the nervous system and the immune system are bidirectional, which is especially important in the impact of psycho-emotional stress on human health [531]. Moreover, chronic psychosocial stress can contribute to the progression of atherosclerosis and its complications [532,533,534].

Perivascular adipose tissue, which produces adipokines and vasoactive substances such as leptin, adiponectin, resistin, and visfatin, cytokines, and growth factors that regulate vascular tone and homeostasis, is also involved in the process of atherogenesis [283,535,536].

The resolution of atherosclerosis can be hindered by an imbalance in the production of pro-inflammatory and specialized pro-resolvent mediators (SPMs), impaired clearance of dead cells, and functional changes in immune cells that contribute to ongoing inflammation [120]. The action of SPMs aims to limit the severity and duration of inflammation, activate regenerative processes, and prevent or reduce post-inflammatory tissue sclerosis. SPMs encompass various endogenous mediators, including non-classical eicosanoids (lipoxins), ω-3 polyunsaturated fatty acid derivatives (resolvins, protectins, and maresins), protein/peptide mediators such as annexin A1, IL-10, IL-37, and nucleotides like adenosine and inosine [120,535].

SPMs have diverse chemical structures and function in a receptor-dependent manner on a wide range of target cells. Most SPMs activate receptors associated with specific types of G proteins. An imbalance between SPMs and pro-inflammatory mediators is linked to several prevalent chronic inflammatory diseases in humans, including atherosclerosis [536]. Consequently, the utilization of SPMs and their chemical derivatives presents a pressing issue in the prevention and treatment of atherosclerosis [120,535].

A stable plaque is characterized by a thick fibrous cap, low inflammatory cell activity, and small necrotic nuclei, which includes foamy cells. It is considered atheroma with predominant I2-dependent productive inflammation with fibrous tissue overgrowth surrounding a stable, moderately sized lipid core. However, excessive fibrous tissue overgrowth, especially in the cap region, can lead to arterial stenosis [536]. Therefore, not all variants of stable plaque development are safe with regard to clinical complications of atherosclerosis.

On the other hand, an unstable plaque is characterized by a dynamic process in the core and the tissues adjacent to the core of the plaque, namely the presence of a necrotic process and incomplete apoptosis of foam cells, and an increased number of pro-inflammatory Mf, T-cells, neutrophils, and other leukocytes, which amplifies the I1 and I3 directions. This leads to an increased role of tissue destruction in the development of inflammation (DAMP formation, among others) but also to thinning of the fibrous capsule, a process that is especially dangerous in the cap region [537,538]. A vulnerable, unstable atherosclerotic plaque is typically characterized as a plaque with a high likelihood of forming a thrombogenic area. An unstable plaque or a plaque prone to ulceration and rupture has a cap less than 65 µm, infiltrated by Mf and T-lymphocytes, a large lipid core constituting more than 40% of the entire plaque area, and increased neovascularization and a large number of frothy cells surrounding the core [539].

Protective angiogenesis plays a crucial role in restoring vascular wall normoxia and resolving inflammation, leading to stabilization and partial regression of atherosclerosis. However, pathological, maladaptive angiogenesis exacerbates disease progression by increasing Mf infiltration and vascular wall thickness and perpetuating hypoxia and necrosis [540]. Additionally, thin-walled, fragile new vessels can rupture, resulting in intra-abdominal hemorrhage.

Several histological types of vulnerable unstable plaques are currently recognized [541,542], including atheroma with a thin fibrous cap (lipid type), plaques with increased inflammation leading to erosion and thrombosis (inflammatory-erosive type), and plaques with necrosis/calcinosis (dystrophic-necrotic type). Hence, among the different types of unstable plaques, there are three distinct types: lipid, inflammatory, and dystrophic-necrotic. Multiple recurrent episodes of rupture, thrombosis, and subsequent fibrosis can occur over several cycles in the same artery section, forming multiple “layers” of fibrous overgrowths.

In general, an unstable plaque’s characteristic feature is the presence of varying degrees of tissue destruction with possible foci of necrosis. The question arises as to the extent the immune system’s response in these cases corresponds to the canons of classical inflammation. The development of classical inflammation in such cases is manifested by several typical variants of exudative-destructive inflammation: purulent, gangrenous, curd-like, and fibrinous [59]. All these variants are characterized by a strong exudative reaction of microvessels, with varying degrees of severity of complement and neutrophil involvement in the process. In the acute course, these inflammation variants usually transform into productive inflammation with granulation formation, followed by potential fibrosis and scar formation. In chronic inflammation, these processes typically develop simultaneously, occupying different compartments of the inflammation focus. Aseptic inflammation, such as autoimmune inflammation, is characterized by the latter variant, related to massive fibrin deposition in the inflammation focus. In some cases, like diabetic kidney disease, fibrinous inflammation may arise as a result of the progression and transformation of local low-grade inflammation [275]. For tissue destruction in atheromatosis, these classical inflammation variants are not observed; rather, atherogenic inflammation, in this case, can be characterized as productive-destructive.

The closest variant of the classic destructive inflammation in atherosclerosis is the rapidly progressing atheroma, characterized by a thin fibrous cap infiltrated with Mf and lymphocytes, rare VSMCs, and underlying necrotic nuclei [543]. Intrabladder hemorrhages with subsequent fibrin deposition in the hemorrhagic zone are common in this type of atheromatosis. However, it cannot be fully attributed to fibrinous inflammation associated with a pronounced exudative reaction of microvessels. The probable reason for these differences is the insufficient inflammatory efficiency of secondary vasa vasorum, which may compensate for the insufficient efficiency of lymphatic drainage of the arterial wall.

**Summary**. The initial manifestations of atherogenesis (pre-atherosclerosis) are characterized by endotheliosis mechanisms typical of low-grade inflammation, while the subclinical stage of lipoidosis (lipid spots and bands) is a transition zone to productive inflammation and atheroma formation. Atherogenic inflammation at the stage of the formed plaque involves a large-scale involvement of the VSMCs as well as vasa vasorum in this process. The formation of a lipid core surrounded by inflammatory cells and pronounced fibrous deposits around this formation is a typical feature of this stage of atheromatosis.

Strengthening of primary and secondary (inflammation-related) alteration factors against the background of efferocytosis and SPM insufficiency leads to increased tissue destruction and unstable plaque formation. However, unlike classical variants of destructive inflammation in atherosclerosis, the vascular-exudative reaction is weakly expressed in this stage of atherogenesis. Therefore, this stage of atherogenesis can be characterized by the position of productive-destructive inflammation but not by the exudative-destructive inflammation of the classical type.

## 7. Relation of Atherosclerosis to Systemic Inflammation (Hyperinflammation)

Acute SI is a pathological process that underlies the development of critical conditions in patients in intensive care units [66,544]. In sepsis, SI is directly associated with rapidly progressing multiple organ dysfunction, which can be complicated by disseminated intravascular coagulation syndrome, acute respiratory distress syndrome, and septic shock [545]. SI, including the “cytokine storm” phenomenon, can also develop in COVID-19 [546]. To verify SI as a general pathological process, certain levels of hypercytokinemia [547] need to be established and compared to the severity of other SI phenomena [66]. The development of acute SI is also typical for the aseptic refractory shock [548] and severe polytrauma variants [509].

The dynamic development of SI is characterized by the alternation of hyperergic and hypoperergic phases, determined primarily by the intensity of its action at the systemic level rather than the nature of the damaging factor [66,549,550]. Chronic SI has less clear clinical equivalents, but its presence can reflect severe chronic diseases with high values of hypercytokinemia and signs of paracoagulation [66]. As applied to atherosclerosis, the relationship between SI and atherosclerosis can be considered from two perspectives: (1) the influence of atherosclerosis complications on the development of acute and chronic SI; and (2) the influence of acute and chronic SI on the development of atherosclerosis.

A study using a combination of existing murine models of atherosclerosis and sepsis found that the intra-abdominal sepsis model accelerated atheroma development [551]. However, the acceleration of atheroma development was associated with prolonged systemic, endothelial, and intimal inflammation and was not explained by ongoing infection. Clinical studies aimed at solving this problem are difficult, and the paucity of clinical data should not be surprising. A literature review by K. Laudanski showed a 3–9% increase in the risk of stroke or acute coronary event in sepsis survivors [552]. One possible explanation for this phenomenon is the persistence of lipid profile abnormalities caused by acute sepsis until recovery, which leads to accelerated atherosclerosis. The presence of an increased risk of cardiovascular disease, including myocardial infarction and stroke, in survivors of sepsis was demonstrated in a review by Merdji et al. [553]. The authors attribute these patterns to premature vascular aging after a critical condition.

In experimental mouse models, Wang et al. (2018) revealed the progression of atherosclerosis after craniocerebral trauma [554]. In humans, posttraumatic stress disorder syndrome has also been found to have an additional influence on atherosclerosis progression [555]. However, the association of this syndrome with SI has not yet been established.

Establishing the relationship between transferred SI and the subsequent progression of atherosclerosis is a significant challenge in acute critical conditions. In chronic diseases, the main challenge is differentiating between SI-dependent systemic inflammatory response and the typical systemic manifestations of classical inflammation and low-grade inflammation. For instance, in gout, classical inflammation (arthritis) develops [556,557], but Kimura et al. (2021) showed in their review that the systemic effects of hyperuricemia contribute to the development of atherosclerosis [558].

We demonstrated a high probability of SI development in systemic autoimmune diseases with established risks of atherosclerosis progression, such as systemic lupus erythematosus, rheumatoid arthritis, and primary antiphospholipid syndrome [66]. However, the influence of autoimmune mechanisms specific to these diseases on atherosclerosis must also be considered. Terminal renal failure, regardless of the features of the primary etiology of renal damage, is a well-known risk factor for the progression of atherosclerosis and its complications, as well as an obvious reason for the development of chronic SI [276].

In general, many diseases with risks of developing chronic SI can contribute to the development of atherosclerosis, such as chronic destructive pulmonary disease and chronic heart failure [558,559,560,561]. However, the problem of determining the relationship between SI and atherosclerosis will largely be determined by the existence of generally accepted and specified theoretical ideas about SI as a general pathological process and the availability of standard approaches to the verification of SI. Nonetheless, it can be concluded that the influence of SI and atherosclerosis on each other is mutual (Figure 5).

The relationship between atherosclerosis and acute systemic inflammation (SI) is well established. Complications of atherosclerosis of specific arteries can lead to the development of acute SI, as demonstrated in the clinic and through data on the pathogenesis of cardiogenic shock [284,560,561,562,563]. Our studies have also shown that hemorrhagic stroke in elderly individuals complicated by severe coma and multiple organ failure is associated with the development of SI, including the phenomenon of paracoagulation (D-dimer > 500 ng/mL) [564]. In turn, we recorded the presence of chronic SI criteria, including paracoagulation, in patients with chronic limb-threatening ischemia caused by common femoral artery atherosclerotic lesions [66]. It should be noted that approximately 6% of adults worldwide have problems related to atherosclerotic ischemia of the lower extremities [565]. Furthermore, we detected signs of chronic SI in some patients over 70 years old who had New York Heart Association (NYHA) functional class II-IV heart failure [566]. While the causes of possible SI development, in this case, are polyvalent, ischemic changes in the heart and other organs related to atherosclerosis are undoubtedly contributing factors. Therefore, atherosclerosis is closely associated with chronic systemic and local low-grade inflammation, and critical complications of atherosclerotic arteries of the brain, heart, and lower extremities can provoke the development of acute and chronic variants of systemic hyperinflammation. In turn, acute and chronic SI can contribute to the progression of atherosclerosis in many diseases.

## 8. Conclusions

In modern medicine, a rational classification of pathologies is necessary not only based on the clinical principle but also on the main characteristics of pathogenesis manifested in various diseases, with their regularities reflecting abstract models of general pathological processes. A fundamental question remains regarding the definition of atherosclerosis as a clinical definition or as a general pathological process involving many universal mechanisms of pathologies while still having an independent meaning. ICD-11 demonstrates that atherosclerosis is a common platform for a large number of clinical definitions rather than a clinical nosology. Pathologies directly related to atherosclerosis occupy several sections in ICD-11, including cerebrovascular diseases, coronary heart disease, and atherosclerotic chronic arterial occlusive disease, all of which have arterial atheroma at the basis of pathogenesis or a typical pathological process leading to its formation.

Initially, atherosclerosis was not considered an inflammatory disease. However, many authors now view atherosclerosis as a low-grade chronic inflammation, which is not entirely accurate, as atheroma formation more closely resembles classical variants of productive inflammation, involving various immune response vectors in the process. Nevertheless, vascular atheroma does not fully align with canonical inflammation due to the low pro-inflammatory activity of vasa vasorum recruited into the damaged intima and the peculiarities of metabolic (cholesterol-dependent) damage. Moreover, despite the possible involvement of autoimmune mechanisms and infectious factors in atherosclerosis pathogenesis, leukocyte participation in productive inflammation is more limited compared to classical autoimmune and infectious diseases.

The primary participants in inflammatory atherogenesis are Mf interacting with cholesterol and modified VSMCs. The multivariate and large-scale transdifferentiation of VSMCs is also a characteristic feature of atherosclerosis. Possibly for these reasons, atherosclerosis is not defined in current literature as vasculitis or arteritis, as other vascular diseases with canonical inflammation are categorized. Instead, atherosclerosis corresponds more closely to noncanonical or quasi-inflammation, which differs from low-grade inflammation in several aspects. These features of atherosclerosis cannot be fully attributed to metabolic damage factors because, in diabetic kidney disease and nonalcoholic fatty liver disease, local low-grade chronic inflammation can transform into classical inflammation variants [59], but this is not entirely characteristic of atherosclerosis. Furthermore, arthritis in gout is characterized by classical inflammation variants, as uric acid functions as both a metabolite and a classical DAMP [539,566,567].

In general, the inflammatory process in atherosclerosis is closely related and shares many common mechanisms with low-grade inflammation (particularly at the initial stages of atherogenesis), as well as with productive inflammation of the classical type at the stage of formed atheroma. Simultaneously, the atherogenic process in unstable plaques can be characterized as productive-destructive inflammation, which is also atypical for destructive-exudative variants of classical inflammation.

Atherosclerosis complications, such as heart attacks and strokes, can lead to the development of acute systemic hyperinflammation. In contrast, complications like critical atherosclerotic ischemia of the lower extremities can result in chronic systemic inflammation. This relationship is bidirectional, as both acute and chronic variants of systemic inflammation can contribute to the progression of atherogenesis.

Therefore, while the atherogenic inflammatory process is related to other types of inflammation, it is not entirely identical to them. It’s important to note that atherosclerosis is a common platform for many significant diseases. Additionally, it is crucial to improve and implement new pathogenetic prevention and therapy methods for atherosclerosis in medical practice [14,568,569,570,571,572]. Advanced methods for studying atherosclerosis in vitro should also be introduced [572]. All of these factors lead us to characterize atherosclerosis as a special variant of the general pathological process.

If models of general pathological processes serve as pathogenetic platforms for many clinical definitions, is there a common platform for general pathological processes themselves? We propose that pro-inflammatory tissue stress is such a common platform, and its elementary functional unit is cellular stress. It’s important to note that the presence of alteration and cellular response to alteration is a condition for survival or programmed cell death not only in pathology but also in physiological conditions [59,60,61]. Moreover, typical cellular stress programs manifest themselves in pathologies not directly related to inflammation, such as tumor growth [59].

This fact highlights the presence of common molecular mechanisms, including epigenetic changes and backbone inducible signaling pathways, in processes that differ in nature and function. However, the context of these mechanisms at the cellular, tissue, and organismal levels gives rise to different functional systems that characterize the models of key general pathological processes (Figure 6).

Section 3 illustrated the link between atherogenesis and all the main universal mechanisms of pro-inflammatory cellular stress. Additionally, atherosclerosis is characterized by typical response mechanisms of the immune system, connective tissue, and vascular endothelium, which are involved to varying degrees in the development of low-grade inflammation and inflammation of the classical type (Section 4 and Section 5). All of this not only differentiates various general pathological processes but also establishes their interrelation. Atherosclerosis may occupy a unique position within this system (Figure 6).

Thus, we believe there are several reasons to view the typical integrative regularities of atherosclerosis pathogenesis as a special variant of the general pathological inflammation process with distinctive features from both classical (canonical) inflammation and local low-grade inflammation.

## Figures and Tables

**Figure 1 ijms-24-07910-f001:**
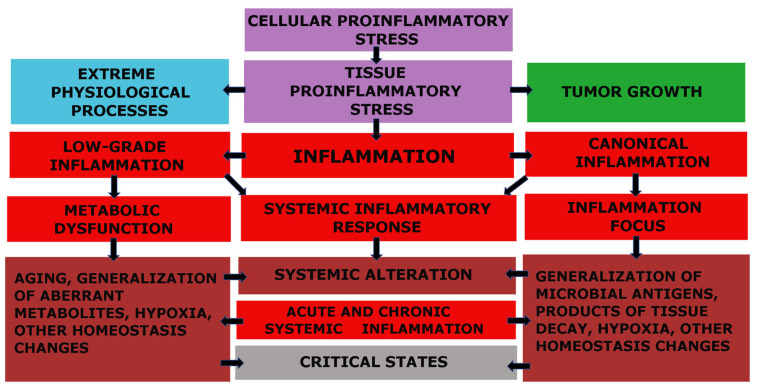
Relationship between cellular and tissue stress and human pathology [66]. **Note:** *The figure shows the relationship between cellular and tissue stress and human pathology. Cellular pro-inflammatory stress is a common response to various forms of damage to macromolecules and external stressors. This stress response is the basis for various physiological processes such as muscle contraction, tissue growth, mucosal function, and lymphocyte selection in primary lymphoid organs. However, it also underlies various pathological processes. Tumor cells and the tumor microenvironment are under stress, but tumor growth is not associated with inflammation. Systemic hyperinflammation, on the other hand, is a dysfunctional process that is a complication of other pathological processes that initiate systemic alteration*.

**Figure 2 ijms-24-07910-f002:**
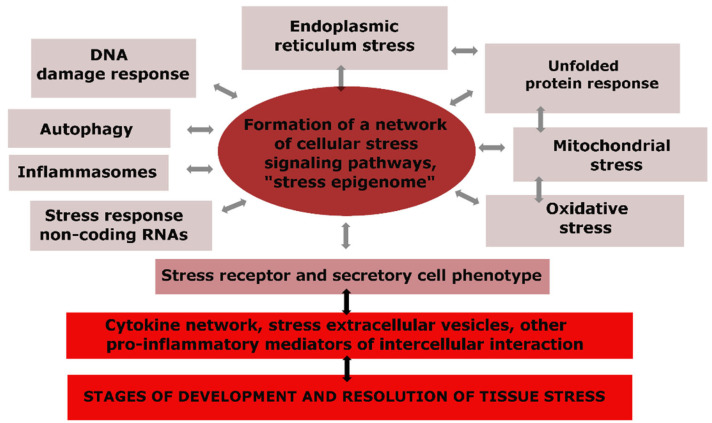
The structure of typical cellular stress processes and their relationship with tissue stress.

**Figure 3 ijms-24-07910-f003:**
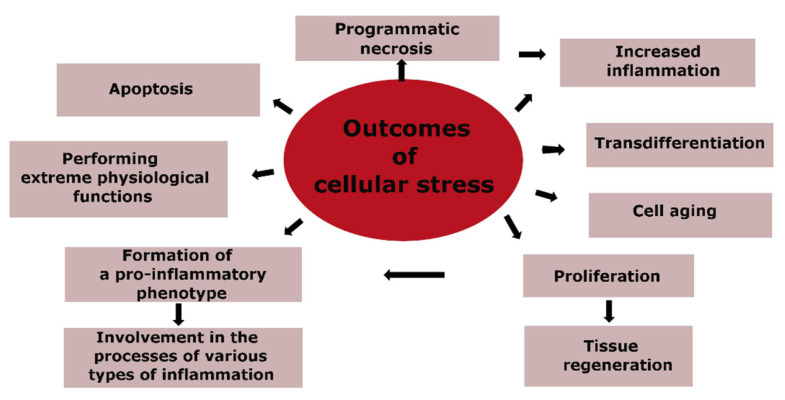
Outcomes of cellular stress.

**Figure 4 ijms-24-07910-f004:**
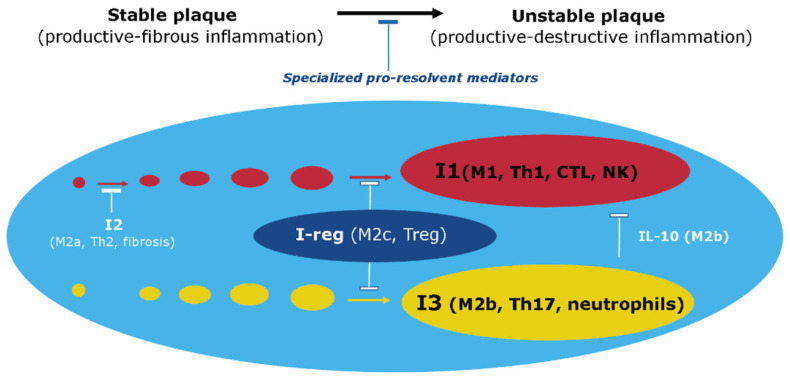
Involvement of Immune Response Vectors in Atherosclerotic Inflammation. ***Note:***
*This figure illustrates the roles of various immune response vectors (I1, I2, I3, and I-reg) in the processes of atherosclerotic inflammation. (1) The formation of an atherosclerotic plaque and its transition to an unstable state depends on the balance of pro-inflammatory and specialized pro-resolvent mediators such as IL-10. (2) I2 dominance at the stable plaque stage promotes fibrosis and competitive inhibition of I1. However, I1 has a more pronounced pro-inflammatory potential and is an obvious mechanism of tissue destruction. (3) Cellular elements I-reg actively secrete anti-inflammatory mediators such as IL-10 to stabilize the atherosclerotic plaque. (4) Factors I3, along with I1, can be prominently activated in unstable plaque formation, where they promote tissue destruction. Nevertheless, M2b can actively secrete not only pro-inflammatory mediators but also IL-10, thereby limiting not only I1 but also I3, including Th17 and neutrophils* [494].

**Figure 5 ijms-24-07910-f005:**
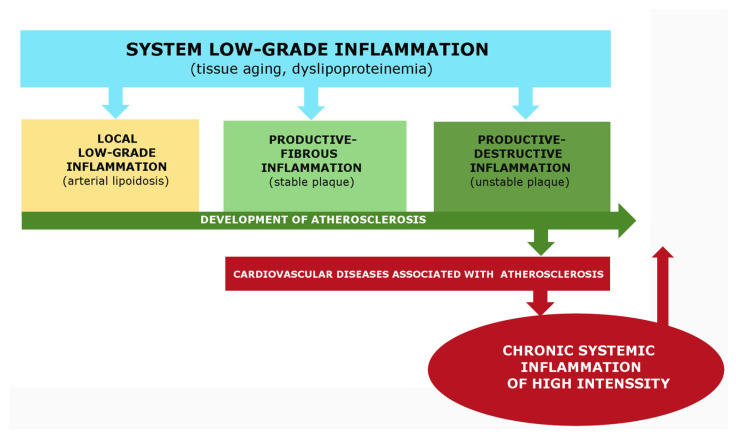
Formation of a Vicious Pathogenetic Circle Integrating Atherogenesis and its Critical Complications. **Note:** This figure illustrates the development of a vicious pathogenetic circle integrating atherogenesis and its critical complications associated with acute and chronic systemic hyperinflammation. (1) The development of atherosclerosis is closely related to tissue aging, metabolic allostasis, and systemic and local low-grade inflammation. (2) Complications related to atherosclerosis can lead to systemic alteration, which is essential for the development of systemic hyperinflammation. (3) The development of acute or chronic systemic inflammation leads to more pronounced changes in homeostasis than low-grade inflammation, including in arteries that are problematic in relation to atherosclerosis, which in turn increases the likelihood and severity of critical complications. (4) In sepsis, severe trauma, various shock conditions, as well as in systemic autoimmune diseases, end-stage renal disease, and some other severe chronic diseases, systemic hyperinflammation may occur primarily in relation to progressive atherosclerosis. However, in this case, systemic inflammation will also provoke the accelerated development of atherosclerosis and the subsequent formation of a negative feedback loop.

**Figure 6 ijms-24-07910-f006:**
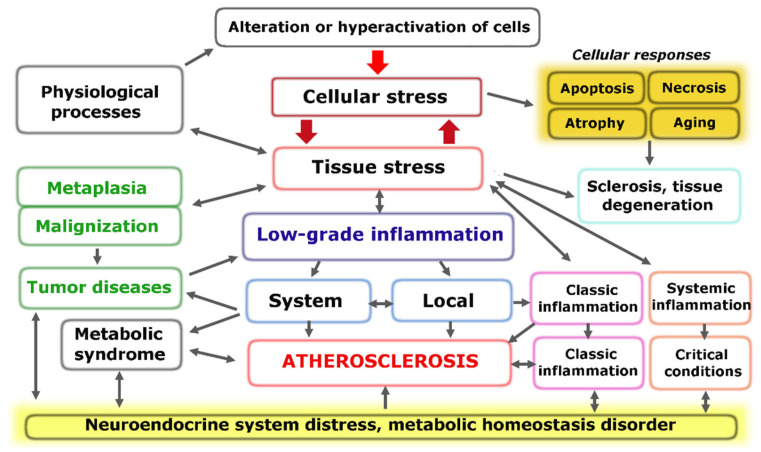
Interrelation of the main general pathological processes on the basis of common mechanisms of cellular and tissue pro-inflammatory stress.

**Table 1 ijms-24-07910-t001:** Scavenger receptors that may be involved in the pathogenesis of atherosclerosis.

Name	Cells	Main Ligands	References
SR-A1 (CD204, SCARA1)	Mf, EC, VSMC	HSP, AGE, PAMP, DNA, oxLDL, acLDL, polyanions, calciproteins, spectrin	[206,213,214,215,216,217]
SR-A5 (SCARA5)	EC	PAMP, oxLDL, acLDL, polyanions, ferritin	[216,218,219,220,221]
SR-A6 (MARCO)	Мf	PAMP, oxLDL, acLDL, polyanions	[215,216,219,222,223]
SR-B1 (CD36L1)	Мf, EC	HDL, oxHDL, oxLDL, acLDL, PAMP, polyanions, carotenoids	[224,225,226]
SR-B2 (CD36)	Мf, EC, VSMC, platelets	Phosphatidylserine, oxLDL, AGE, thrombospondin, PAMP, polyanions, FFA, collagen, fibronectin, integrins	[219,227,228,229,230]
SR-D1 (CD68)	Мf, monocytes	oxLDL, E-selectins, phosphatidylserine, oxLDL	[206,221,231,232]
SR-E1 (LOX-1)	EC, VSMC, Мf, platelets	oxLDL, AGE, CRP, apoptosis products, HSP, activated platelets	[206,221,233,234,235]
SR-E3 (CD206, MR-1)	Мf	Microbial glycans, altered plasma glycoproteins, collagen, HSP70	[236,237,238,239,240]
SR-F1 (SREC-I)	EC, Мf	oxLDL, acLDL, calcium-binding proteins (advillin, calretinulin), PAMP, HSP, C1q	[206,221,235,241,242]
SR-F2 (SREC-II)	EC, Мf	acLDL, C1q, calreticulin	[243]
SR-G1 (CXCL16)	Мf, VSMC, platelets	oxLDL, PAMP, products of apoptosis and cell necrosis, CXCR6	[219,221,244,245]
SR-H1 (Stabilin-1)	Мf	PAMP, oxLDL, acLDL, heparin, hyaluronate, phosphatidylserine	[246,247,248]
SR-H2 (Stabilin-2)	Мf, EC	PAMP, oxLDL, acLDL, acid glycans, nucleic acids, phosphatidylserine	[247,249]
SR-I1 (CD163)	Мf, monocytes	Complexes hemoglobin-haptoglobin, fibronectin, PAMP	[250,251,252,253]
SR-J1 (RAGE)	Мf, EC, VSMC	AGE, phosphatidylserine, oxLDL, DAMP (S-100, HMGB1), HSP70, Mac-1 (CD11b/CD18)	[221,254,255,256]
SR-K1 (CD44)	Leukocytes, Мf	Hyaluronate, collagens, fibronectin, laminin, E-selectins, fibrin, osteopontin, metalloproteinases	[257,258,259]
SR-L1 (CD91, LRP1)	VSMC, Mf, monocytes	Apolipoprotein E, oxLDL, proteinase-antiproteinase complexes, HSP, C1q, integrins, thrombospondins, fibronectin, lactoferrin, PDGF	[260,261,262]

Note. Cells: Mf—macrophages, EC—endothelial cells, VSMC—vascular smooth muscle cells. Ligands: PAMP—pathogen-associated molecular patterns, HSP—heat shock proteins (from the families: HSP70, HSP90, HSP110), AGE—Advanced Glycation End products, FFA—free fatty acids, C1q—complement system factor, acLDL—acetylated low-density lipoproteins (LDL), oxLDL/acLDL—oxidized/acetylated LDL, HDL—high-density lipoprotein, CXCR6—chemokine receptor, CRP—C-reactive protein, HMGB1—nuclear non-histone protein with DAMP function, S-100—calcium-binding proteins with DAMP function, Mac-1 (CD11b/CD18) — beta-2-integrin, PDGF—platelet-derived growth factor.

**Table 2 ijms-24-07910-t002:** Vectors of immune response (I) [460,461,462,463,464,465,466,467,468,469,470].

I	Th (TFs), Cytokines: Activators and *-Inhibitors	Main Cytokines Th	Other Cells (TFs; Cytokines Production // Reception; *-Inhibitors)	Major Role in Inflammation	Complications
I1	Th1(T-bet, STAT4, STAT1); IL-12, IFN-γ; IL-4 *, IL-10 *	IFN-γ, IL-2, CXCL10, CXCL11	M1 (STAT1, NF-κB, IRF9; TNF-α, IL-1β, IL-6, IL-12, IL-15, IL-23, CXCL9// IFN-γ, TNF-α; IL-10 *, TGF-β *), CTL, NK, ILC1 (IFN-γ)	Response to intracellular infection, antitumor immunity	Autoimmune processes, allograft rejection
I2	Th2 (GATA3, STAT5, STAT6); IL-4, IL-25, IL-33; IFN-γ *, TGF-β *, IL-12 *	IL-4, IL-5, IL-13, IL-25, CCL17, CCL22	M2a (STAT6, STAT3, GATA3, PPAR; IL-6, IL-10, CCL17 // IL-4, IL-13, IL-33), Tc2 (IL-5, IL-13), mast cells, basophils, ILC2 (IL-4), epithelial cells, eosinophils	Antimetazoan immunity, chronic inflammation, inflammation in damage-sensitive tissues	Allergic processes, I1 suppression, tissue fibrosis
I3	Th17 (RORγt, RORα, STAT3, STAT5); IL-1β, IL-6, IL-23, TGFβ; IL-10*	IL-17A/F, IL-21, IL-22, CCL20, CXCL-1,7,20	M2b (NF-kB, IRF3; TNF-α, IL-1β, IL-6, IL-10, CCL1 // IL-17A/F, TNF-α, IL-1, IL-6, IL-23; IL-10 *), Tc17 (IL-17), neutrophils, ILC3	Response to extracellular infection	Autoimmune processes, allograft rejection
I3	Th22 (RUNX3, AHR, STAT3); IL-6, IL-1β, TNF-α	IL-22, CCL-2, 20, CXCL-9, 10, 11, FGF	Epithelial cells, Langerhans cells	Protection of the epidermis against extracellular infection	Autoimmune skin processes
Ireg	Treg (FOXP3, STAT3/5, SMAD2/3, RORγt GATA3,); IL-2, IL-10, TGF-β	TGFβ, IL-10, CCL4	M-reg, M2c (SMAD2, SMAD3, STAT3; IL-10, TGFβ, CXCL13 // IL-10, TGF-β), Tr1 (IL-10, IFN-γ), Tc-reg (TGFβ, IL-10), ILC10 (IL-10)	Limiting the expression of I1 and i3, inhibition of the autoimmune process	I1 and I3 immunosuppression

Note: *- Inhibitors of immune response; TFs—Transcription factors (with the main TFs underlined); Th—CD4+ T-helper; CTL—Cytotoxic T lymphocytes, or Tc1; NK—Natural killer cells; Tc—CD8+ T cells; Treg—CD4+ regulatory T cells; ILC—Innate lymphoid cells; Tr1—Type 1 regulatory T cells (CD4+). Some authors categorize Th9 as well, which are induced by TGF-β and IL-4 from Th2 precursors (the main TF is PU.1). Th9 cells are major producers of IL-9, contribute to anti-tumor immunity (in contrast to Th2), but may also participate in autoimmune processes [466,467,468,469,470,471,472,473,474,475]. Pro-inflammatory Th-GMs are the primary T-cells producing GM-CSF. Simultaneously, Th-GMs actively produce IL-2, TNF-α, IL-3, and CCL20, and when T-bet expression increases, they can also actively secrete IFN-γ [476].

## Data Availability

No new experimental data were created.

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
