# Peer review of "Atherosclerosis and Inflammation: Insights from the Theory of General Pathological Processes"

_ijms, 2023, doi:10.3390/ijms24097910_

Round 1
Reviewer 1 Report
This is a well structured review examining the pathophysiology of atherosclerosis and the role of inflammation.
The authors give a long and comprehensive overview of the pathophysiological mechanisms underlying the development and outcome of atherosclerosis, emphasizing the contribution of the inflammatory response in this process. They suggest atherosclerosis should be considered as a one of a kind inflammation.
The review covers the main relevant literature in the field. I would recommend the authors to consider the vaste and accruing evidence that signifies
A) that inflammation is actively resolved by specific endogenous mediators
B) that such endogenous proresolving mediators may be defective in atherosclerosis, leading to perpetuation of what they call “productive-destructive inflammation”
C) that harnessing endogenous proresolving mediators and their downstream pathways offer beneficial opportunities for treating cardiovascular dieseases
Author Response
Dear Reviewer,
We would like to express our sincere gratitude for your insightful comments and suggestions on our manuscript titled "Atherosclerosis and Inflammation: Insights from the Theory of General Pathological Processes". Your feedback has been invaluable in improving the overall quality of our work. We have carefully considered your comments and have made the necessary changes and amendments to our manuscript accordingly. Below, we address each of your concerns and outline the changes made in response.
Reviewer 1:
- A) In response to your suggestion, we have expanded our discussion on the role of endogenous mediators in the resolution of inflammation in the context of atherosclerosis. We have rewritten a new section and have incorporated relevant literature to emphasize their importance in this process.
- B) We agree that the defective proresolving mediators in atherosclerosis are critical and have therefore included a discussion on this topic in the newly added section. We have highlighted the role of these mediators in perpetuating the "productive-destructive inflammation" process in atherosclerosis.
The action of SPMs aims to limit the severity and duration of inflammation, activate regenerative processes, and prevent or reduce post-inflammatory tissue sclerosis. SPMs encompass various endogenous mediators, including non-classical eicosanoids (lipoxins), ω-3 polyunsaturated fatty acid derivatives (resolvins, protectins, and maresins), protein/peptide mediators such as annexin A1, IL-10, IL-37, and nucleotides like adenosine and inosine [543, 544].
SPMs have diverse chemical structures and function in a receptor-dependent manner on a wide range of target cells. Most SPMs activate receptors associated with specific types of G proteins. An imbalance between SPMs and pro-inflammatory mediators is linked to several prevalent chronic inflammatory diseases in humans, including atherosclerosis [544]. Consequently, the utilization of SPMs and their chemical derivatives presents a pressing issue in the prevention and treatment of atherosclerosis [543, 544].
We believe that the changes made to our manuscript have significantly improved its quality and readability. Once again, we thank you for your valuable comments and suggestions, which have greatly contributed to the enhancement of our work. We hope that our revisions will meet your approval and look forward to your further feedback.
Sincerely, the authors.
Reviewer 2 Report
The review compiles the main mechanisms related to atherosclerosis. It describes risk factors, stages, and specially its relation to classical and non-classical mechanisms of inflammation, which is the main goal of the review. It is an important review for the field since it link several aspects of the disease.
I would like to suggest the correction of the line 42 "HDL – липопротеины высокой плотности" and the review of the English.
I would suggest improve figures 4 and 5 or present a more informative legend to the figures. A image should be easy to understand even if the reader haven't read the text.
Other wise, the manuscript seems good and achieves the aim and its objectives.
Author Response
Dear Reviewer,
We would like to express our sincere gratitude for your insightful comments and suggestions on our manuscript titled "Atherosclerosis and Inflammation: Insights from the Theory of General Pathological Processes". Your feedback has been invaluable in improving the overall quality of our work. We have carefully considered your comments and have made the necessary changes and amendments to our manuscript accordingly. Below, we address each of your concerns and outline the changes made in response.
Reviewer 2:
We apologize for the error on line 42 and have now corrected it to "HDL - High-density lipoproteins."
We have thoroughly reviewed the manuscript and revised the language (almost in the Abstract) to improve clarity and coherence. We have also sought the assistance of a native English speaker to ensure the quality of the language used in the manuscript.
In response to your suggestions, we have redrawn and improved the quality of Figures to ensure that they are easy to understand and convey their intended message effectively. Additionally, we have revised the legends for these figures to provide a more informative description.
We believe that the changes made to our manuscript have significantly improved its quality and readability. Once again, we thank you for your valuable comments and suggestions, which have greatly contributed to the enhancement of our work. We hope that our revisions will meet your approval and look forward to your further feedback.
Sincerely, the authors.